



# Mechanisms and predictability of Sudden Stratospheric Warming in winter 2018

Irene Erner[1,2], Alexey Yu. Karpechko[1], Heikki J. Järvinen[2]

[1]Finnish Meteorological Institute, 00560 Helsinki, Finland

[2]Institute for Atmospheric and Earth System Research, Faculty of Science, University of Helsinki, 00014 Helsinki, Finland

**Correspondence**: Irene Erner (irene.erner@fmi.fi)

**Abstract.** In the beginning of February 2018 a rapid deceleration of the westerly circulation in the
polar Northern Hemisphere stratosphere took place and on 12 February the zonal mean zonal wind at
60° N and 10 hPa reversed to easterly in a Sudden Stratospheric Warming (SSW) event. We
investigate the role of the tropospheric forcing in the occurrence of the SSW, its predictability and
teleconnection with the Madden-Julian oscillation (MJO) by analysing the European Centre for
Medium-Range Weather Forecasts (ECMWF) ensemble forecast. The SSW was preceded by
significant synoptic wave activity over the Pacific and Atlantic basins, which led to the upward
propagation of wave packets and resulted in the amplification of a stratospheric wavenumber 2
planetary wave. The dynamical and statistical analyses indicate that the main tropospheric forcing
resulted from an anticyclonic Rossby wave breaking, subsequent blocking and upward wave
propagation in the Ural Mountains region, in agreement with some previous studies. The ensemble
members which predicted the wind reversal, also reasonably reproduced this chain of events, from
the horizontal propagation of individual wave packets to upward wave activity fluxes and the
amplification of wavenumber 2. On the other hand, the ensemble members which failed to predict
the wind reversal, also failed to properly capture the blocking event in the key region of the Urals and
the associated intensification of upward propagating wave activity. Finally, a composite analysis
suggests that teleconnections associated with the record-breaking MJO phase 6 observed in the late
January 2018 likely played a role in triggering this SSW event.

## 1 Introduction

Sudden stratospheric warmings (SSWs) are the most prominent phenomena taking place in the
wintertime polar stratosphere and representing the dynamical linkage between troposphere and
stratosphere. During a major SSW event the zonal mean zonal winds at 10 hPa and 60° N reverse
from westerlies to easterlies and the stratospheric temperature rises by several tens of Kelvins over
the course of a few days (Butler et al., 2015). SSWs have been shown to be related to the enhancement
of tropospheric forced planetary wave packets that propagate upward into the stratosphere and interact



with the mean flow (Charney and Drazin, 1961; Matsuno, 1971; McIntyre, 1982; Limpasuvan et al.,

2004). These upward propagating planetary waves amplify with height, approaching the critical level where they irreversibly break and deposit westward angular momentum (quantified as a convergence of the Eliassen-Palm flux), which leads to the deceleration and breaking down of the polar night jet (Polvani and Saravanan, 2000). Stratospheric circulation anomalies, in turn, can influence the troposphere (Kuroda and Kodera, 1999; Baldwin and Dunkerton, 1999). In particular, it can lead to

the development of negative phase of the Northern Annular Mode (NAM), shifting tropospheric storm tracks southward and making northern and central Europe prone to cold Arctic air masses (Thompson et al., 2002). SSWs occur approximately once every second winter; however, there is no regularity, and during the last decade the events occurred in 2013, 2018, 2019.  The 2013 and 2018 events were followed by cold and snowy weather in Europe (Nath et al., 2016; Karpechko et al.,

2018). Since the stratosphere tends to be more predictable than the troposphere, SSWs are considered to be a potential source of extended-range predictability (Christiansen, 2005; Sigmond et al., 2013; Scaife et al., 2016; Karpechko, 2015; Domeisen, Butler, Charlton-Perez, et al., 2019; Kautz *et al.*, 2019). It is therefore important to understand factors controlling the variability of the polar vortex and SSWs generation.

External forcings such as the quasi-biennial oscillation (QBO) (Holton and Tan, 1980), Madden-Julian oscillation (MJO) (Garfinkel et al., 2012) or El Niño Southern Oscillation (ENSO) (Taguchi and Hartmann, 2006; Song and Son, 2018) may trigger such anomalous stratospheric states as SSWs acting as a source of Rossby wave packets or influencing their vertical propagation (Lu et al., 2012). It has been shown that some major SSWs have been preceded by tropospheric blockings that modify

tropospheric planetary waves in such a way that they can influence the onset and type of an SSW (Nishii and Nakamura, 2004; Martius et al., 2009; Woollings et al., 2010; Castanheira and Barriopedro, 2010; Quiroz, 1986).

The onset and dynamical evolution of each SSW event is a combination of the typical characteristics and its unique features, therefore detailed investigation of each case can advance our

understanding of large-scale processes in the boreal winter stratosphere and improve their prediction. On 12 February 2018 a prominent vortex split type major SSW occurred (hereafter referred to as SSW2018) (Karpechko et al., 2018; Lee et al., 2019). The split type events are considered to be less predictable than the displacement events, especially at lead times of 1–2 weeks (Domeisen et al., 2019). SSW2018 occurred under the favourable conditions of the easterly phase of QBO, La Niña

phase of ENSO and followed the MJO phase 6 with the largest amplitude in observational record (from 1974 to 2018) (Barrett, 2019). Barrett (2019) showed that the large-amplitude MJO episode in 2018 affected weather in the north-eastern United States under the conditions of strengthened Rossby





wave teleconnections between the tropics and the extratropics. Furthermore, SSW2018 was preceded by a record-breaking meridional eddy heat flux at 100 hPa observed before an SSW since 1958 (see
Fig. A1 in Appendix A).

In this study we investigate the role of the tropospheric forcing in SSW2018, its predictability and teleconnection with the MJO by analysing the European Centre for Medium-Range Weather Forecasts (ECMWF) ensemble forecast. The purpose of the paper is to present results of the analysis of the atmospheric circulation in the stratosphere and troposphere before and during SSW2018 and
clarify the driving mechanisms focusing on the amplification of the upward wave activity propagation into the stratosphere before the SSW onset. Karpechko et al. (2018) showed that the lead time for the SSW2018 prediction varied among the 11 individual models of the subseasonal-to-seasonal (S2S) database of extended range forecasts. They suggested that the errors in the forecast location of an anticyclone over the Urals (the Ural high) played the crucial role in reducing the SSW2018
predictability. This result is being proved in the present study with additional analysis of the Ural high onset. The importance of wave breaking in the building of the Ural high and critical role of an Atlantic cyclogenesis was highlighted by Lee et al. (2019). On the other hand, Rao et al. (2018) pointed to the Alaskan blocking as the source of intensified extratropical wavenumber 2 planetary wave that was important for triggering SSW2018. In this paper we will extend the analysis of previous
papers and present further evidence that several Rossby wave trains that developed in the troposphere and originated from localized quasi-stationary blocking highs have likely contributed to the SSW2018 forcing.

The paper is organized as follows. In Section 2 the data and analysis methods are briefly described. In Section 3 we present dynamical features of SSW2018 and contrast evolution of forecast ensemble
members that predicted and did not predict SSW2018 at 11 days lead time. In particular, we present evidence that MJO teleconnection played a role in triggering SSW2018. In the final section we present our conclusions.

## 2   Data and Methods

This study is based on the ECMWF 46-day coupled ocean-atmosphere ensemble forecast , produced
twice a week (Tuesday and Thursday) with 51 members (Vitart et al., 2017). We chose the forecast initialized on 1 February 2018 to test the predictability of SSW2018 and analyse the error growth. The date is selected based on Karpechko et al. (2018) and Lee et al. (2019) who showed that this was the first forecast date when a considerable fraction of ensemble members predicted SSW2018. To discern the errors and their possible sources, we selected two groups of ensemble members for further
analysis and comparison with the reanalysis fields:





- EN+ cluster: 10 ensemble members which succeeded in forecasting the wind reversal at 10 hPa and 60° N within +/– 1 day from the observed onset date (12 February) (Fig. 1);

- EN– cluster: 10 ensemble members which maintained the largest positive values of the zonal mean zonal wind at 10 hPa and 60° N across the ensemble members.

Hereafter, we analyse the composite fields of these two groups while all ensemble members are used to illustrate forecast spread and correlations for several diagnostics.

For the forecast verification, we use the ECMWF ERA-Interim reanalysis (ERA-I, Dee et al., 2011). The present analysis includes the period from 1979 to 2018. 12-hourly data are used on a 1°×1° horizontal grid covering the Northern Hemisphere (NH).

Stratospheric wind, eddy heat flux and wave activity flux are analysed as full fields while geopotential height is analysed as an anomaly. ERA-I anomalies are calculated with respect to the period 1980–2010. The forecast anomalies are defined as the subtraction of the model's own climatology from the forecast fields. Model's own climatology is computed using hindcasts over the prior 20 years: 1997–2017.

The wave activity flux (WAF) that indicates a propagating packet of planetary waves in the three-dimensional space can be used to localize regions on wave activity sources and sinks. Following Plumb (1985), wave activity flux $\overrightarrow{F_s}$ on the sphere is represented in log-pressure coordinates as:

$$\overrightarrow{F_s} = \begin{pmatrix} F_x \\ F_y \\ F_z \end{pmatrix} = p\,cos\varphi \begin{pmatrix} v'^2 - \dfrac{1}{2\Omega a\,sin2\varphi} \dfrac{\delta(v'\Phi')}{\delta\lambda} \\ -u'v' + \dfrac{1}{2\Omega a\,sin2\varphi} \dfrac{\delta(u'\Phi')}{\delta\lambda} \\ \dfrac{2\Omega\,sin\varphi}{S}\left[v'T' - \dfrac{1}{2\Omega a\,sin2\varphi} \dfrac{\delta(T'\Phi')}{\delta\lambda}\right] \end{pmatrix}, \tag{1}$$

where $F_x, F_y, F_z$ denote the zonal, meridional and vertical components of the wave activity flux respectively; $p$ is pressure, $\varphi$ and $\lambda$ are latitude and longitude respectively, $u$ and $v$ are zonal and meridional winds, $\Omega$ is the Earth's rotation rate, $a$ is the Earth's radius, $\Phi$ is geopotential, $T$ is temperature and $S$ is the static stability parameter. The prime denotes perturbations from the zonal mean values.

The Madden–Julian oscillation (MJO) phase is determined using the seasonally independent Real-time Multivariate MJO index (RMM). It is based on time series of the two leading principal components derived from empirical orthogonal functions (EOFs) of the combined fields of near-equatorially averaged 850 hPa zonal wind, 200 hPa zonal wind, and satellite-observed outgoing longwave radiation (OLR) data (Wheeler and Hendon, 2004). The RMM index is divided into eight phases that broadly correspond to the regions of enhanced convection.



## 3 Results

### 3.1 Stratospheric forecasts

We start by analysing the predictability of SSW2018 in the ECMWF ensemble forecast. Figure 1 shows the temporal evolution of the observed and forecasted zonal mean zonal wind at 10 hPa and 60° N (U10) for individual ensemble members during February 2018. In the forecast initialized on 1 February, there is a weak SSW signal: 14 ensemble members (~27 %, orange dashed lines) predicted wind reversal within 1 day from the observed onset date. The forecasted SSW probability, defined as a fraction of ensemble members predicting an SSW at each day (Karpechko, 2018; Taguchi, 2016; Tripathi et al., 2016), was 0.06 on the observed onset date of 12 February and increased to 0.31 by 14 February when the minimum values of U10 were achieved by most ensemble members. The spread of predicted wind speed among the members increases markedly after 9 February when the observed polar night jet underwent the strongest deceleration. The fluctuations of the easterlies observed in the reanalysis after the reversal are not captured by any ensemble members. Karpechko et al. (2018) showed that most ensemble members underestimated the eddy heat flux at 100 hPa which is used to characterize the upward planetary wave propagation from the troposphere to the stratosphere since it is proportional to the vertical group velocity of a planetary wave and to the vertical component of the Eliassen-Palm flux (Newman et al., 2001).

The evolution of zonal mean zonal winds at 10 hPa for February 2018 is shown in Fig. 2. Early in the month, the axis of the polar night jet is located at around 70° N and shifting gradually poleward (Fig. 2a). On 11 February, the jet quickly decelerates around 80° N and the zonal wind reversal occurs in high latitudes and extends from the North Pole to about 50° N. Easterly wind peaks of – 30 m/s are found on 12–16 February and around 21 February after diminishing to zero on 17 February at 60° N. The northward shift of the polar night jet occurs prior to the zonal wind reversal – a feature highlighted in some previous SSW studies and pointed out as a precondition for the effective wave forcing because in this case the relatively small mass and moment of inertia of the vortex allow upward propagating waves to distort it (Limpasuvan et al., 2004; McIntyre, 1982; Harada et al., 2010; Nishii et al., 2009). Overall, easterly winds dominate the polar stratosphere north of 50° N from mid-February to March.

The composite of the EN+ members (Fig. 2b) captures well the northward shift of the polar jet axis in the beginning of the February and the wind reversal on 12 February. The composite mean underestimates the magnitude and duration of easterlies, recovering to westerly flow after 18 February. This could possibly reduce the forecasted impacts of SSW2018 on surface. Several



ensemble members, however, maintained easterlies until the end of February matching the magnitude of the observed easterlies (Fig. 1). The EN– composite (Fig. 2c) maintains westerlies throughout February.

Figure 3 shows that in the beginning of February, the centre of the polar vortex is already displaced from the pole towards Greenland and Norwegian sea and a high over the Alaska begins to develop. During 4–6 February the two troughs over Northern America and Central Siberia and the anticyclone over Alaska start to form. By 7–9 February another high over the North Atlantic begins to develop (wavenumber 2 planetary wave pattern, Fig. 3a). During 10–12 February the two highs merge over

the pole leading to a vortex split. The low over Canada intensifies while the other part of the split vortex weakens over Siberia, leading to the circulation reversal at 60° N (Fig. 3d). To reveal forecast errors we compare the EN+ and EN– members composites to the reanalysis (Fig. 3 b,c,e,f). Analysis shows that during the first ~7 days after the initialization the forecast errors in the stratosphere are modest, consistent with the analysis of Karpechko (2018), but they start to grow after 7 February

mainly near the position of one of the daughter vortices over Northern America in both EN+ and EN– clusters (Fig. 3 b, c). By 10–12 February, the EN– cluster notably underestimates the magnitude of the merged high that had replaced the polar vortex over the pole, and it shows bigger errors in the position of the cyclone over Canada (Fig. 3f) compared to the EN+ cluster (Fig. 3e). However, the overall structure of the errors appears remarkably similar in the two groups which might suggest the

presence of a systematic model bias.

Long planetary waves are known to interact with the mean flow before SSWs (e.g. Limpasuvan et al., 2004). Time evolution of the planetary waves amplitudes in the beginning of the February 2018 is shown in Fig. 4. The highest wave activity in the NH stratosphere is concentrated within the latitudinal range of 40° N – 75° N (e.g. Peters et al., 2010), therefore this belt of latitudes was chosen

for averaging. Planetary wave with wavenumber 1 (PW1) dominates in the beginning of February in the middle stratosphere, but its amplitude decreases rapidly and reaches its minimum on 10 February. On the other hand, the amplitude of PW2 starts to grow rapidly on 4 February reaching values of 90 dam on 10 February just before the SSW2018 onset (Fig. 4a). Such inverse correlation of these two planetary waves is often observed before major split type SSWs as the propagation characteristics of

the waves differ depending on the zonal wavenumber and wave period (Charney and Drazin, 1961). A strong PW2 increase often results in a vortex splitting (McIntyre, 1982), as it happened in February 2018. Figures 4b and c depict the time evolution on the first three waves for the each of the 10 chosen EN+ and EN– members. First, the evolution across individual ensemble members in both categories is remarkably similar, though the spread in the EN– cluster is bigger. The overall evolution pattern in

the EN+ cluster resembles well the ERA-I verification (Fig. 4b). The EN– members fail to capture





the amplitude growth of the PW1 after 10 February and, in addition to that, they underestimate the PW2 amplitude (Fig. 4c). PW3 remains weak in both observations and forecast ensembles.

On 7 February the polar vortex had already been weakened and distorted (Fig. 3a) and the polar night jet started to decelerate. Horizontal distribution of the ensemble spread in the lower stratosphere

is shown in Fig. 5. The largest ensemble spread is mainly confined to the subpolar North Atlantic (Fig. 5a) where the forecast errors on that date are the largest (Fig. 3c). Throughout the period of vortex deceleration the area of the large forecast spread at 50 hPa height gradually expands horizontally and, by 12 February, it covers most of the polar stratosphere north of 70° N (Fig. 5b,c).

To better understand sources of ensemble spread in the stratosphere we look at the zonal cross-

sections (Fig. 6). As seen in Fig. 6a, there are three areas of large ensemble-forecast spread on 7 February, when the polar vortex started to decelerate and be distorted: over the Ural Mountains, Alaska and North Atlantic regions. The areas with the large spread extend from the troposphere into the lower stratosphere. Blocking anticyclones in these regions were pointed out to be associated with SSW tropospheric forcing (Martius et al., 2009; Woollings et al., 2010; Rao et al. (2018); Karpechko

et al., 2018), as they may act as the source of the Rossby-wave packets that propagate into the stratosphere and lead to a SSW onset. The upward group velocity propagation of the waves is indicated by the westward tilt of the geopotential anomaly lines with height (Fig. 6). The spread can be explained by the inconsistencies in the location, amplitudes and group velocities predicted by different ensemble members (Nishii and Nakamura, 2010). In fact, most of the ensemble members

started to underestimate the heat flux entering the stratosphere (Fig. 1 in Karpechko et al., 2018) after 7 February.

To further analyse contribution of these three regions to the SSW2018 forcing we examine the timeseries of the vertical component of wave activity flux averaged zonally and over the three continuous longitudinal ranges. The main wave event is identifiable in lower and middle stratosphere

prior to the circulation reversal (Fig. 7a), preceded by the upward flux maxima in the lower and mid-troposphere on 4 February with the time lag of ~7 days needed for the planetary wave to propagate vertically from the troposphere to the stratosphere. The division into three longitudinal ranges allows us to investigate the wave activity flux propagation between the troposphere and the lower and middle stratosphere over the limited longitudinal ranges (Harada et al., 2010; Coy and Pawson, 2015). The

North Atlantic third (Fig. 7j) shows the biggest maxima of vertical wave activity flux in the troposphere in the beginning of February and also in the lower and middle stratosphere just before the SSW onset compared to the other two thirds of the globe. The similar propagation pattern can be seen in the Europe/Siberia third (Fig. 7d). The North Pacific third (Fig. 7g) shows an increased upward flux before the event which is restricted to the lower stratosphere.





Comparison of the similar diagnostics of vertical WAF performed for the EN+ and EN–
composites (Fig. 7, 2 and 3 column respectively) shows that the EN+ cluster captures the wave
propagation patterns zonally averaged (Fig. 7b) and in all three longitudinal ranges (Fig. 7 e, h, k)
although it somewhat underestimates the magnitudes of fluxes. The EN– forecasts composite does
not predict a significant vertical wave propagation from the troposphere into the stratosphere in either
of the longitudinal ranges.

        While the EN– forecasts failed to reproduce the increases in wave activity flux in all three regions,
it is not clear where the errors were crucial for the failed SSW forecast. The correlation analysis of
the zonal mean WAF at each level averaged over 4–11 February with forecast U10 on 12 February
across individual ensemble members shows the negative correlation at all levels, starting from the
lower troposphere, at 0.05 significance level (Fig. 8a). The correlation coefficient increases with
height reaching r = –0.95 at 50 hPa. When split into the three regions, the wave activity contributions
from the Siberia and North Atlantic sectors are significant in the lower and middle stratosphere, with
strongest correlations found in the Siberian sector (Fig. 8b). This suggests that upward wave activity
propagation in these regions was critical for the SSW2018 forcing. On the other hand, the correlation
analysis shows that there is no significant relation between WAF in the North Pacific sector and U10.

### 3.2   Tropospheric waves

We next look at the tropospheric precursors of SSW2018. The three areas with the largest forecast
spread (Fig. 6) are associated with blocking ridges seen in the 250 hPa geopotential height (Fig. 9a).
Several wave packets manifested as meandering westerlies can be distinguished in the consecutive
geopotential height fields over the period of 3–9 February. Most pronounced one is associated with
the anticyclonic wave breaking episode over the North Atlantic also demonstrated by Lee et al.
(2019). Here, a well-developed ridge can be seen on 3 February. During 4–6 February this ridge
breaks anticyclonically propagating downstream until blocked by the developing anticyclonic ridge
over the Ural region around 6–7 February. The second ridge can be distinguished developing over
eastern cost of North America on 5 February and propagating downstream during 5–8 February. At
the same time a stationary upper troposphere ridge is seen over Alaska over the whole period.

        The propagation of the synoptic features can be also diagnosed using the squared meridional wind
fields (Nishii and Nakamura, 2010). Figure 9b shows that, between 3 and 7 February, the maximum
of the squared 250 hPa meridional wind propagated across the North Atlantic and Northern Eurasia
with an average group speed of ~27° in longitude per day before being blocked over the Urals with
little downstream propagation thereafter. Such propagation speed is consistent with group velocity of





baroclinic waves (Chang, 1993; Nishii and Nakamura, 2010) suggesting that formation of the blocking anticyclone was the result of a downstream development. Figure 9b also shows that the stationary Alaskan ridge served as a source of two more individual wave packets that propagated

towards the Atlantic starting on 3 and 6 February respectively.

The EN+ composite of the squared meridional wind at 250 hPa (Fig. 10a) is in agreement with the reanalysis (Fig. 9b), capturing all three wave packets discussed above, whereas in the EN– cluster the wave train over the Ural region disappears starting from 6 February. Thus, the wave packet associated with the Ural blocking fades away in the EN– members. Although the propagation of other wave

packets is captured by the EN– cluster, there are differences with respect to the EN+ cluster in the location and magnitude of the packets. In particular the magnitude of the Atlantic ridge on 6–9 February is strongly underestimated. The differences between the EN+ and EN– clusters can also be investigated by looking at the forecast spread in the meridional wind at 250 hPa that represent inconsistency among the ensemble members (Fig. 10c). Downstream propagation of the forecast

spread is well distinguishable in the Fig. 10c and it is strongly associated with the propagation of the aforementioned wave packets. Interestingly, there is large spread also in the eastern Pacific associated with the quasi-stationary Alaskan ridge.

To see the behaviour of the wave packets in more detail we studied the horizontal propagation of WAF (Plumb, 1985). The observed wave activity in the mid-troposphere and the difference between

the EN+ and EN– clusters are shown in Fig. 11 for 5–7 February. We focus on this time period because this is when large differences between these two ensembles have emerged and we use 3-day averaging following previous practices of using this diagnostic developed for quasi-stationary waves (e.g. Harada et al., 2010; Peters et al., 2010). The 500 hPa pressure level is chosen to highlight the mid-tropospheric processes. The same diagnostics in the upper troposphere (300 hPa) yield similar

results (not shown). Figure 11a shows eastward propagation of wave activity along the jet stream in the reanalysis, with large values seeing in all three regions of anomalous highs identified in the previous sections. The eastward wave activity propagation is stronger in the EN+ members through most of the NH extratropics with the greatest differences following the meandering extratropical jet stream (Fig. 11b). Remarkable difference in the horizontal propagation of wave packets is seen over

all three centres of forecast uncertainty discussed above – Alaskan, North Atlantic and Ural suggesting underestimation of eastward wave activity propagation in the EN– cluster. To inspect closely the difference in wave propagation between the EN+ and EN– clusters we look at the magnitude of the horizontal wave flux within the areas representative for these regions marked in Fig. 11b as two boxes (over the Ural region (Box 1) and the North Atlantic (Box 2)). Over the North

Pacific, since the anomalous flux change its direction within the area, we choose to analyse the flux





through the two surface lines (Fig. 12). The wave activity propagation over the Box 1 in the EN+ cluster captures well the sharp amplification seen in the ERA-I verification between 5–9 February. This amplification corresponds to the period of the development of the Ural blocking high (Fig. 12a). The EN– cluster fails to capture this intensification of the wave activity. The wave activity fluxes

through the surfaces defined by the Lines 1 and 2 (Fig. 12b,c) are comparable between the EN+ and EN– clusters and reproduce the fluctuations seen in ERA-I, but in general they underestimate the observed values. The analysis of the net flux in the North Atlantic region (Box 2) does not show significant differences between the ensemble members and ERA-I: the three individual peaks between 4 and 10 February, corresponding to the three individual wave packets revealed in Fig. 9–

10, are well captured in the forecast (Fig. 12d). Thus, results in Fig. 12 suggest that the key difference between the EN+ and EN– forecasts in terms of horizontal wave activity propagation is in the Ural region.

To demonstrate that the differences in horizontal WAF in the mid-troposphere between the EN+ and EN– clusters are relevant for SSW forecasting we perform the correlation analysis of zonal mean

zonal wind in the mid-stratosphere and the zonal component of WAF at 500 hPa across all ensemble members. Over the Northern Siberia, the correlation field (Fig. 13) resembles the location of the biggest differences in WAF between the EN+ and EN– clusters (Fig. 11) with statistically significant negative correlation coefficients exceeding –0.5. Thus, the negative correlation in the Ural region indicates that the stronger flux in the region is associated with weaker stratospheric winds and

suggests that errors in the wave activity in the location of the Ural high turn out to be crucial for forecasting SSW2018, consistent with the results by Karpechko et al. (2018) and Lee et al. (2019).

### 3.3    Teleconnection with MJO

In the 10-day period before the SSW2018 central date (27 January–7 February) an active MJO in phases 6 and 7 with large amplitude prevailed in tropical Indian ocean and South China Sea (Barrett,

2019). It has been shown that MJO phase 6/7 events associated with OLR anomalies in Eastern Pacific can lead to weakening of the polar vortex through enhancement of upward propagating wave fluxes towards Alaska and are often followed by SSWs (Schwartz and Garfinkel, 2017). In this section we assess the evidence that the MJO played a role in the onset of SSW2018. We chose for the analysis the ensemble forecast initialized on the 1 February and, as the amplification of the MJO phase 6

occurred prior to that date, it is expected that the wave activity source associated with MJO has been included into forecast initial conditions, potentially leading to the more precise forecast of SSW2018. We find no evident link between the skill of MJO forecast and SSW2018: the EN+ members do not



predict MJO more correctly that the EN– members (see Fig. A2 in Appendix A). Based on that we focus on analysis of MJO teleconnections, testing the hypothesis that correct forecasting of MJO

teleconnections was important factor in simulating SSW2018.

To verify that, first, we constructed the composite field of geopotential height anomalies picked only for days with MJO phase 6 with the lag of 5–9 days in both ECMWF historical forecasts and ERA-I. It is very difficult to clearly establish the causality between tropical oscillations and polar anomalies, because of the complex interactions between the propagating waves and the mean flow.

Therefore, one of the ways to approach causality is to use time lag. The lag of 5–9 days after MJO phase 6, which took place on 27–31 January, roughly corresponds to the period in the early February when tropospheric waves forced SSW2018 based on the analysis in the previous sections. In particular, the ridge over the North Atlantic was developing during this period. This suggests that the MJO phase 6 fingerprint should be taken with the lag of 5–9 days.

We start by testing how well can the model reproduce MJO phase 6 teleconnection in the extratropics. In Fig. 14a the composite fields showing the observed fingerprint of the MJO phase 6 are presented with contours. Figures 14 b and c show similar fingerprint but constructed with the model hindcasts over 20 years. These two fields both have prominent lows in the North Pacific and over Canada and highs over the Ural, western North America and the North Atlantic. Although the

fingerprint fields show some dissimilarity in the positions and strength of the features, their overall structure is well captured by the model. This result is in line with Vitart (2014, 2017) who showed that the model produces realistic patterns of MJO teleconnections.

Figure 14a also shows the observed geopotential anomalies field averaged for 5–7 February, i.e. 5–7 days after the end of the MJO phase 6. Although the spatial correlation between the two fields is

small (r=0.03), likely because the key features in 2018 are somewhat displaced with respect to the climatological composite,  the overall structure of the field prior to SSW2018 strongly resembles the climatological MJO response, capturing the anomalous highs over the Siberia, high-latitude Pacific and North Atlantic, as well as the low over the Canada. On the other hand, the low in the North Pacific region is not pronounced and the high over western America is displaced towards northwest.

Although the evidence is not conclusive, analysis of Fig. 14a support the idea that MJO teleconnections may have played a significant role in dynamical evolution of the extratropical atmosphere preceding SSW2018, consistent with existing literature (Schwartz and Garfinkel, 2017).

The composite field made for 5-7 February 2018 using the EN+ members captures the observed structure of geopotential height field well with PW2 pattern prevailing in the northern latitudes, and

also strongly resembling the MJO fingerprint composite (Fig. 14b). On the contrary, the response in the EN– cluster does not resemble either the observed field or the MJO composite and shows a PW1





pattern with two highs in Alaska and Ural region merged together (Fig. 14c), which is consistent with the EN– forecasts not capturing the amplification of PW2 in the stratosphere (Fig. 4c). In summary, our composite analysis provides supportive, although not decisive, evidence that teleconnections

associated with MJO phase 6 played a role in triggering SSW 2018 both in observations and in the forecasts.

## 4    Discussion and Conclusions

Using the ECMWF ensemble forecast we examined the predictability of the major SSW in the middle of February 2018. We focused on the identification of the involved dynamical processes and studied

the role of the tropospheric forcing leading to the polar vortex split.

First, we have selected two groups of ensemble members based on the zonal mean zonal wind at 10 hPa and 60° N metric to discern spatial and temporal distribution of forecast errors and its possible sources by comparing the ensemble composites to the reanalysis fields. SSW2018 was preceded by the amplification of PW2 and record-breaking eddy heat flux in the lower stratosphere. This

amplification was reasonably well captured by forecast ensemble members predicting SSW2018 but not those that did not predict it. The forecast error in geopotential height in the mid-stratosphere is small until 7 February and starts to grow mainly near the edge of the polar vortex following its displacement towards North America, marked also by the largest ensemble spread. The growth of the forecast spread was linked to the positions of tropospheric blocking ridges suggesting that their

accurate prediction was important for forecasting the SSW2018 event. The amplification of the stratospheric PW2 was related to a PW2 pattern in the mid-troposphere and was apparently brought about by accumulative effects of localized propagation of wave packets. The period preceding SSW2018 was characterized by the enhanced wave activity in the troposphere. In the Pacific region wave activity fluxes maintained quasi-stationary ridge over Alaska. Over North Atlantic, eastward

propagation of individual wave packets could be identified and tracked back to the Alaskan ridge which served as their source. We show that the propagation of the forecast uncertainties is associated with the downstream propagation of these synoptic patterns in the troposphere, and the subsequent upward propagation of the wave packets to the stratosphere. Comparison of the EN+ and EN– forecast composites reveals that the EN+ forecasts correctly captured the whole chain of the observed events,

from downstream propagation of individual wave packets, to the upward propagation of wave activity, amplification of stratospheric PW2 and breaking down of the stratospheric polar vortex. On the other hand, our analysis suggests that EN– members underestimated both horizontal and vertical WAF propagation. In particular, it is found that the development of the upper troposphere blocking





anticyclone over the Ural region around 6–7 February following the energy injection from wave
breaking over North Atlantic during 4–6 February was largely missing in the EN– cluster. This wave
breaking event was also highlighted by Lee et al., (2019) as being important for amplifying a high-
pressure system over the Urals and triggering SSW2018. Here we also have showed that the crucial
for the Ural blocking wave packet does not appear in the ensemble members that failed to capture
SSW2018. According to our statistical analysis, forecasted stratospheric winds are mostly correlated
with horizontal zonal wave activity flux over the Ural region, with stronger WAF in that region being
associated with weaker stratospheric winds. Furthermore, correlation analysis also reveals that
weaker stratospheric winds in the forecast were mostly associated with the vertical propagation of the
wave activity flux over the Siberian sector with a contribution from the North Atlantic sector. While
we also find enhanced vertical wave activity propagation from the Alaskan sector, correlation analysis
of the forecast members suggests that WAF over this region did not contribute to the SSW2018
forcing, which is somewhat inconsistent with results by Rao et al., (2018), who concluded that
SSW2018 is caused by the Alaskan blocking.

SSW2018 was preceded by the highest ever observed MJO phase 6 which could create favourable
conditions for strengthened Rossby wave teleconnections between the tropics and the extratropics.
We have shown that the anticyclonic centres over the North Atlantic, Ural and Alaska regions formed
before SSW2018 correspond to the MJO phase 6 response pattern taken with the lag of 5–9 days.
These centres were captured well by the EN+ members while the EN– cluster  failed to reproduce the
PW2 structure in the northern latitudes. The composite analysis provides an evidence, although not
decisive, that teleconnections associated with MJO phase 6 played a role in triggering SSW2018.

We conclude by pointing out the importance of the accurate prediction of the strength and position
of synoptic scale mid- and upper tropospheric features and understanding the origin of planetary wave
anomalies for improving the prediction of SSW events. Though the predictability of the 1–2 weeks
for SSW2018 falls within the usual range of predictability for the split events (Karpechko, 2018;
Domeisen et al., 2019), the exceptional conditions before the event could have potentially enhanced
the predictability. It is important to understand what part of the forecast error was associated with
internal variability and what part was due to systematic bias, which is planned to be addressed in a
follow up study.

*Data availability.* All data is available from the European Centre for Medium Range Weather
Forecasting (ECMWF). See https://apps.ecmwf.int/datasets/ for more details. Madden–Julian
oscillation   (MJO)   phase   is   available   from   The   Bureau   of   Meteorology:
http://www.bom.gov.au/climate/mjo/.



*Code availability.* Code is available from Irene Erner upon request.


*Author contributions.* IE performed data analysis and wrote the first draft of the manuscript. AK formed the idea for the study, contributed to the interpretation of the results, and improved the final manuscript. HJ provided guidance on interpreting the results. All authors commented on the manuscript.


*Competing interests.* The authors declare that they have no conflict of interest.

*Acknowledgements.* IE is funded by the Magnus Ehrnrooth foundation and Finnish Meteorological Institute (FMI). AK is funded by the Academy of Finland (grants 286298, 294120, and 319397). This

work is based on (subseasonal-to-seasonal) S2S data. S2S is a joint initiative of the World Weather Research Programme (WWRP) and the World Climate Research Programme (WCRP).





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



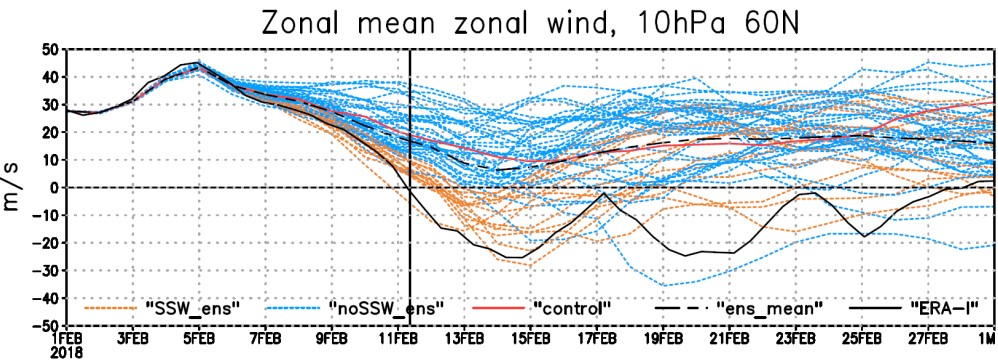


**Figure 1.** Zonal mean zonal wind at 10 hPa and 60° N (m s$^{-1}$). Ensemble forecast initialized on 1 February (orange lines denote ensemble members that predict wind reversal with max 1 day delay, red line – control forecast, black dashed line – ensemble mean) and the ERA-I reanalysis (black solid line). Vertical line denotes the SSW2018 central date.




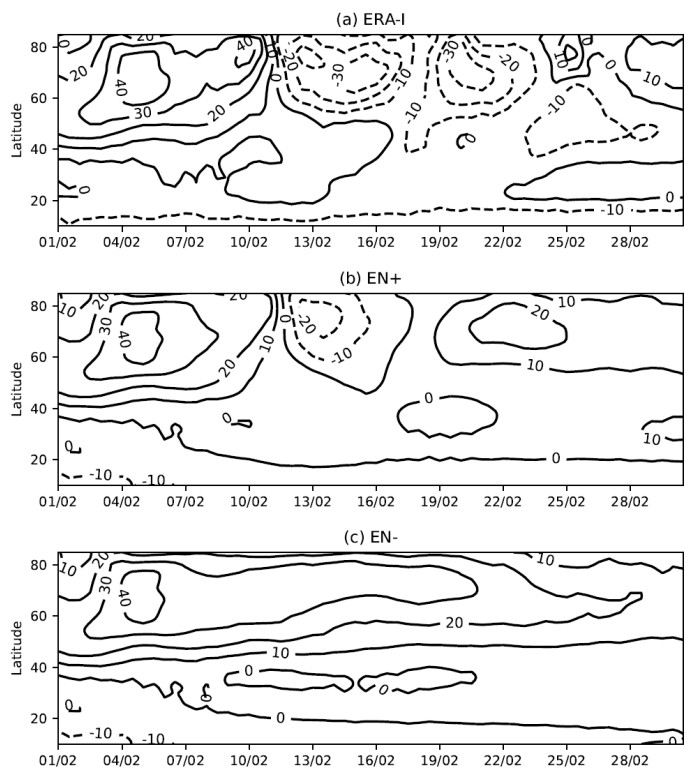

**Figure 2.** Latitude–time cross sections of zonal-mean zonal winds (m s⁻¹) at 10 hPa during February 2018. (a) ERA-I; (b) composite of EN+ members; (c) composite of EN– members. Contour intervals are 10 m s⁻¹.


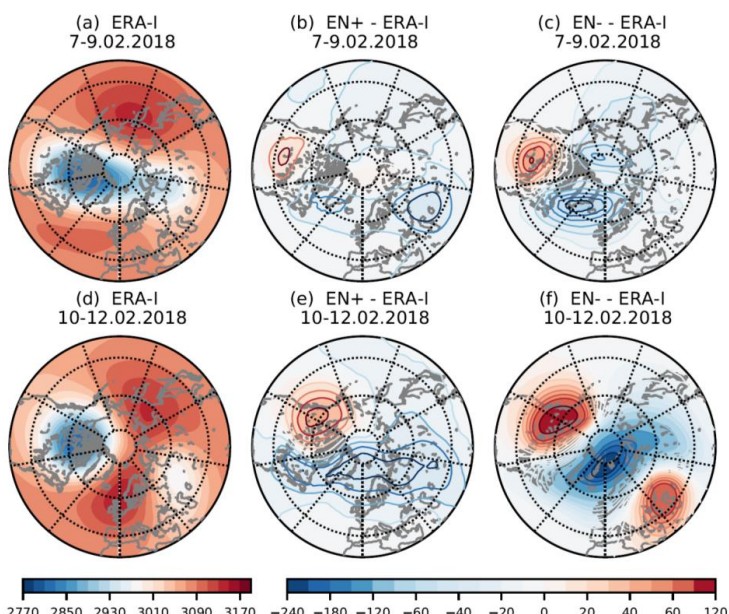

**Figure 3.** Geopotential height at 10 hPa (dam) for two successive 3-day means starting from 7 February (a, d). Difference in geopotential height at 10 hPa (dam) between ERA-I and EN+ members (b, e) and EN– members (c, f).




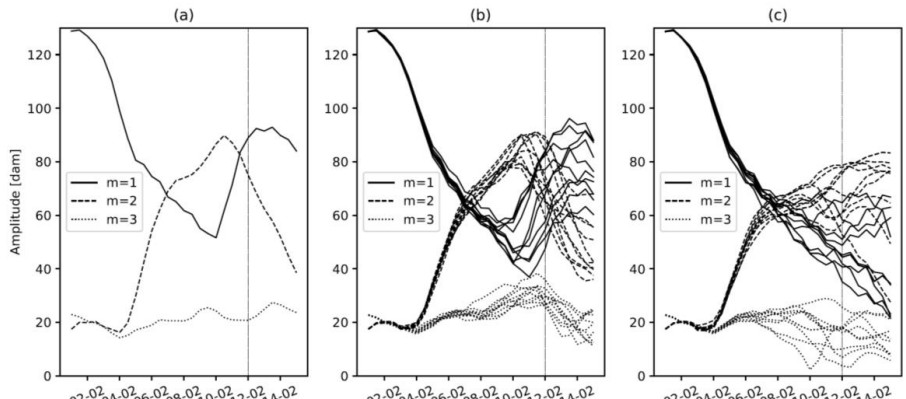

**Figure 4.** Time series of amplitudes of planetary waves with wavenumbers m = 1, 2 and 3 in geopotential height (dam) at 10 hPa averaged over the latitudinal belt 40° N–75° N (a) ERA-I reanalysis, (b) EN+ members, (c) EN– members. Vertical line denotes the SSW2018 central date.




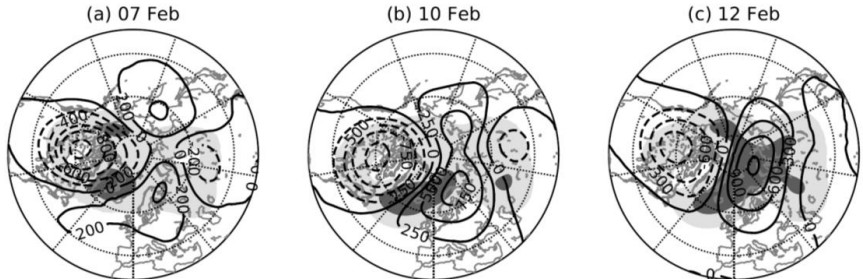

**Figure 5.** ERA-I 50 hPa geopotential height anomalies (contours, m) with respect to the 1980–2010 climatology and ensemble spread of geopotential height predicted for (a) 7, (b) 10 and (c) 12 February 2018 (shaded lightly and heavily for 0.3–0.6 values and values greater than 0.6, respectively). The spread has been normalized by the minimum and maximum values within the domain north of 20° N.




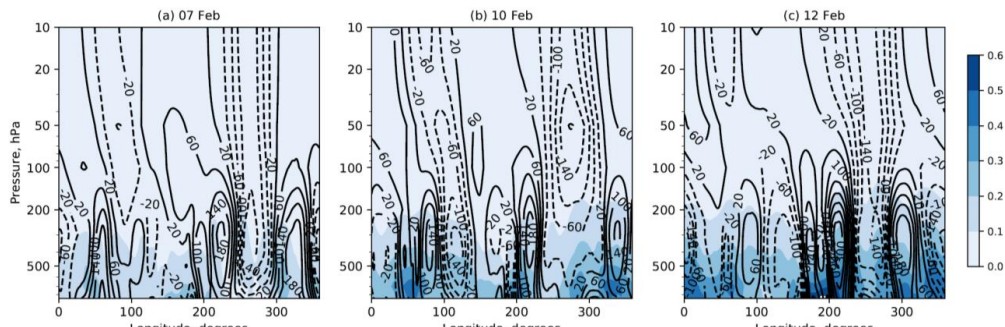

**Figure 6.** Zonal cross-sections for 50° N of the ensemble spread of geopotential height predicted

for (a) 7, (b) 10 and (c) 12 February 2018. Superimposed contours represent observed geopotential anomalies (m) with respect to the 1980–2010 climatology. Solid lines represent anticyclonic (positive) anomalies and dashed lines cyclonic (negative) anomalies. Spread and anomaly are normalized by pressure.



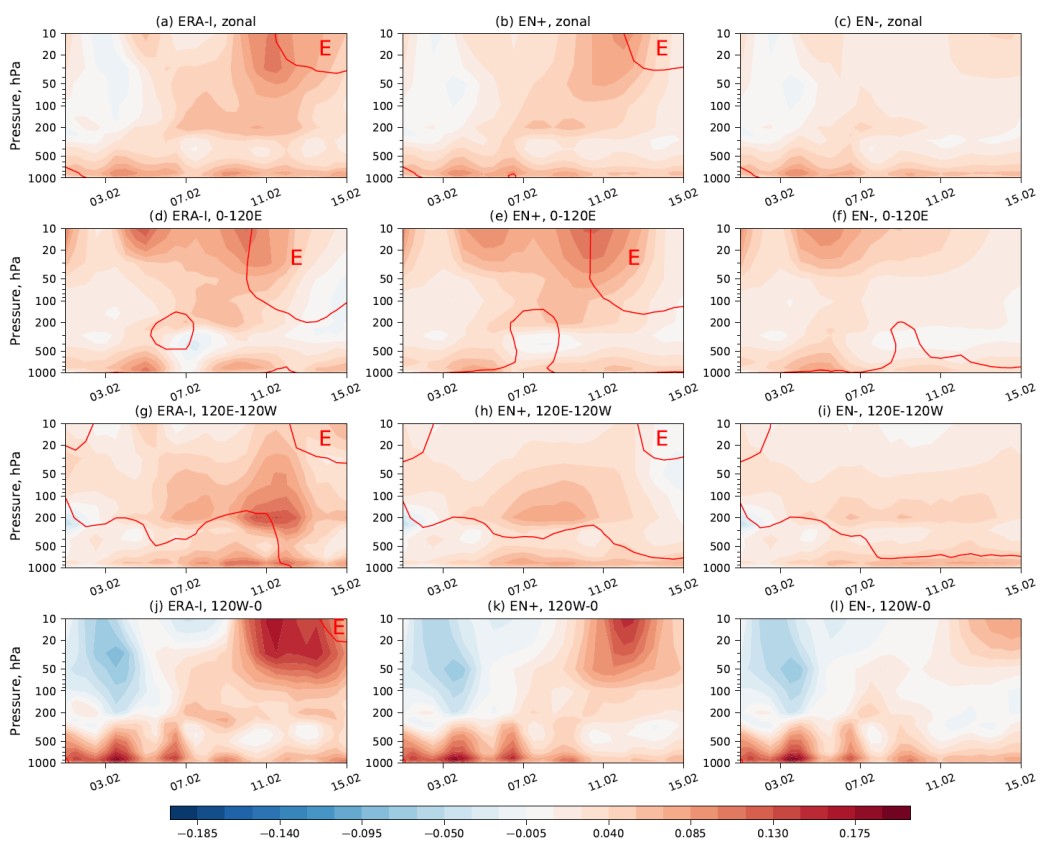

**Figure 7.** Time-altitude plot of the vertical component of WAF (m$^2$ s$^{-2}$, shaded, averaged over 45–90° N, vertically scaled by square root of 1000 hPa/p) and zero zonal wind contour (red) averaged over 55–65° N. (a–c) Zonally averaged, (d–f) averaged over 0°–120° E, (g–i) averaged over 120° E–120° W, (j–i) averaged over 120° W–0°. The red letter 'E' denotes regions of easterly winds. (a, d, g, i) ERA-I; (b, e, h, k) EN+ composite; (c, f, i, l) EN– composite.

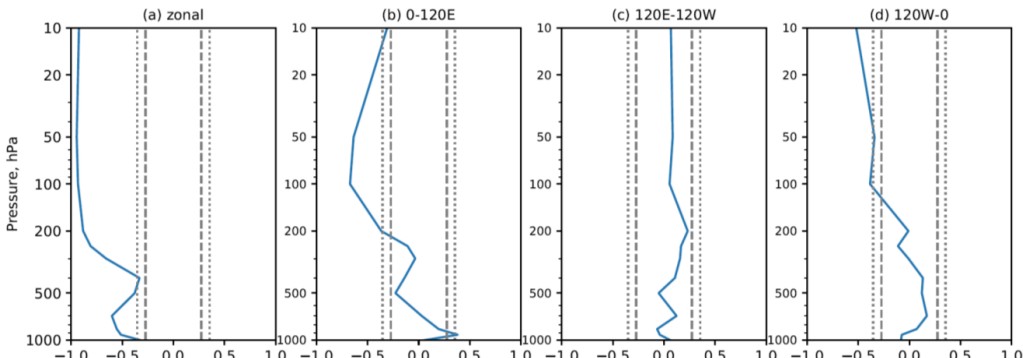

**Figure 8.** Vertical distribution of the correlation coefficient between the vertical component of the WAF forecasts averaged during 4–11 February and U10 forecasts valid on 12 February across individual forecast ensemble members. (a) Zonally averaged; (b) averaged over 0°–120° E; (c) averaged over 120° E–120° W; (d) averaged over 120° W–0°. Dashed vertical lines denote the 0.05 significance level, dotted vertical lines denote the 0.01 significance level.



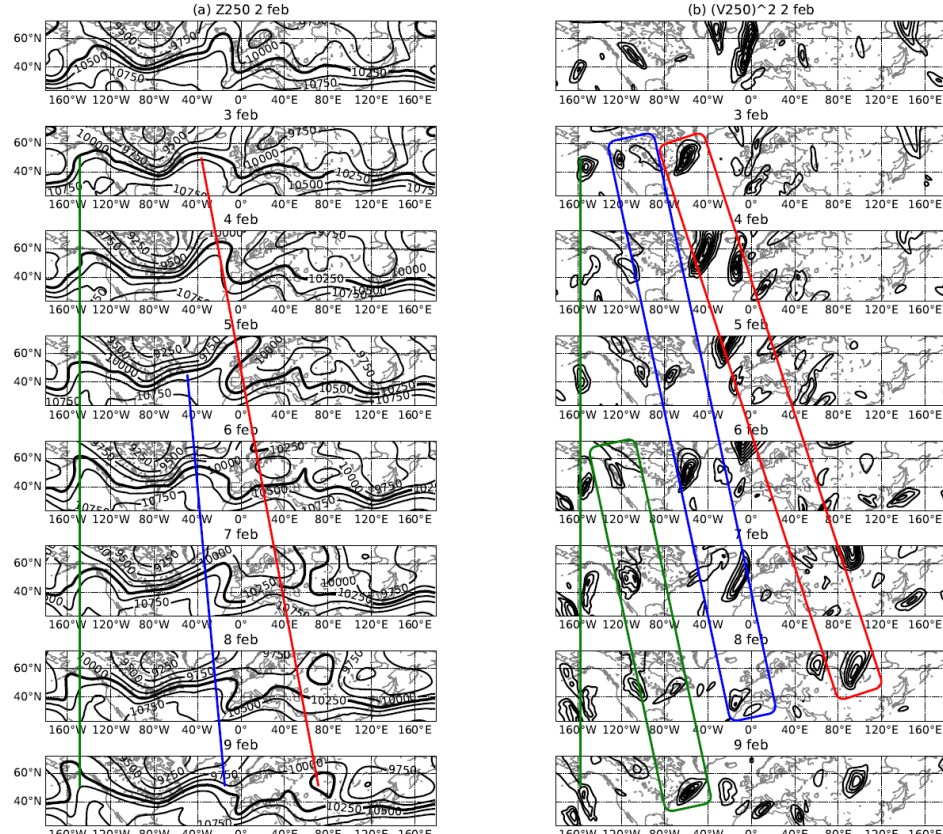

**Figure 9.** Time sequence of (a) ERA-I 250 hPa geopotential height observed from 2 (top) to 9 (bottom) February 2018 over a domain (20° N–70° N). The thick contour corresponds to 10250 m. (b) ERA-I 250 hPa meridional velocity squared.



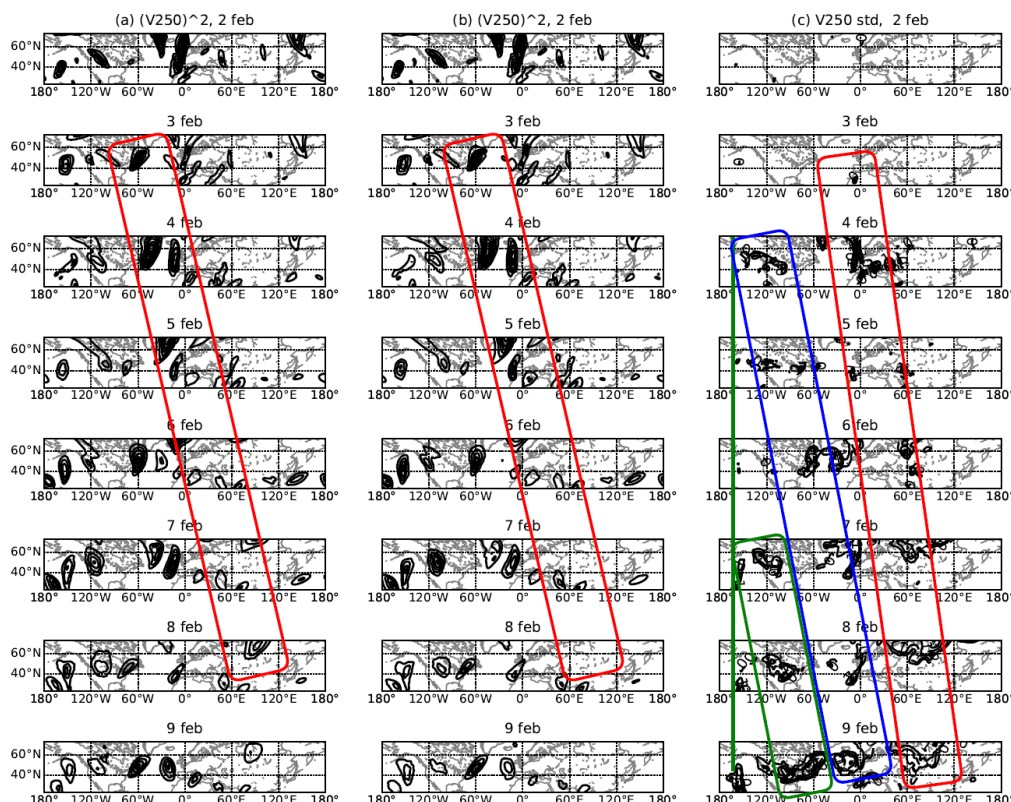

**Figure 10.** Same as Fig. 8b, but for EN+ (a) and EN– (b) members. The red rectangles denote the wave train discussed in text. (c) standard deviation of the predicted 250 hPa meridional wind velocity among ensemble members. The standard deviation is normalized by the maximum and minimum within the domain. Contour intervals are 0.1 starting from 0.5.



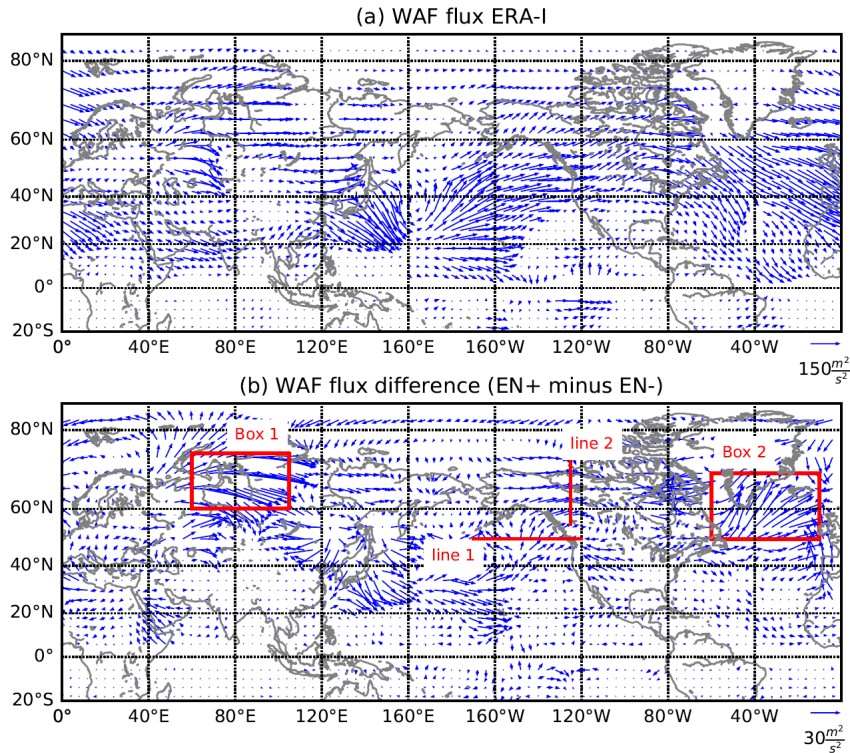

**Figure 11.** The 500 hPa horizontal WAF ($m^2 \, s^{-2}$) averaged over 5–7 February. (a) ERA-Interim; (b) difference between EN+ and EN− groups of ensemble members.



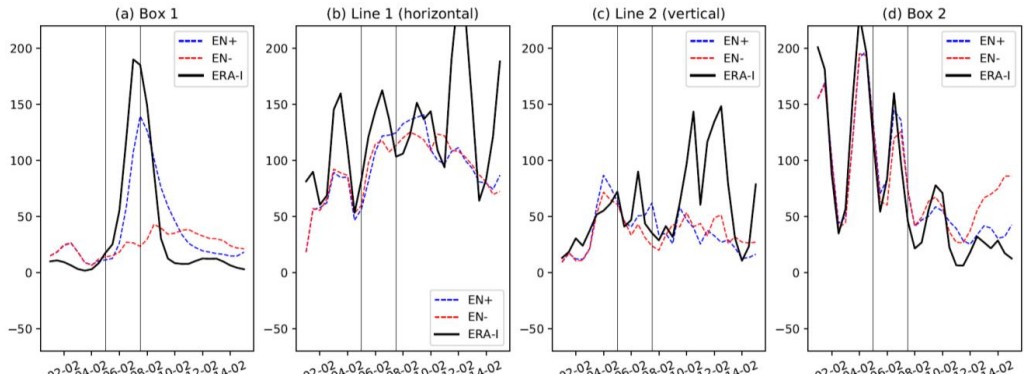

**Figure 12.** Time series of the horizontal WAF at 500 hPa ($m^2\,s^{-2}$) averaged over the Boxes 1 and 2 and through the Lines 1 and 2 shown in Fig. 11b. (a) and (d) show mean length of the horizontal WAF vector while (b) and (c) show mean meridional and zonal components respectively. Grey vertical lines denote the averaging period taken for analysis in Fig. 11: 5 and 7 February.

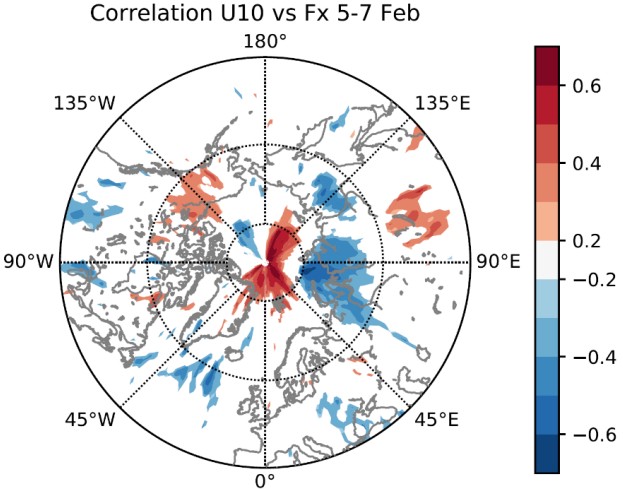

665

**Figure 13.** Correlation coefficient between zonal WAF at 500 hPa averaged 5–7 February and U10 reanalysis on 12 February across individual ensemble members. All shaded coefficients are significant at p = 0.05.

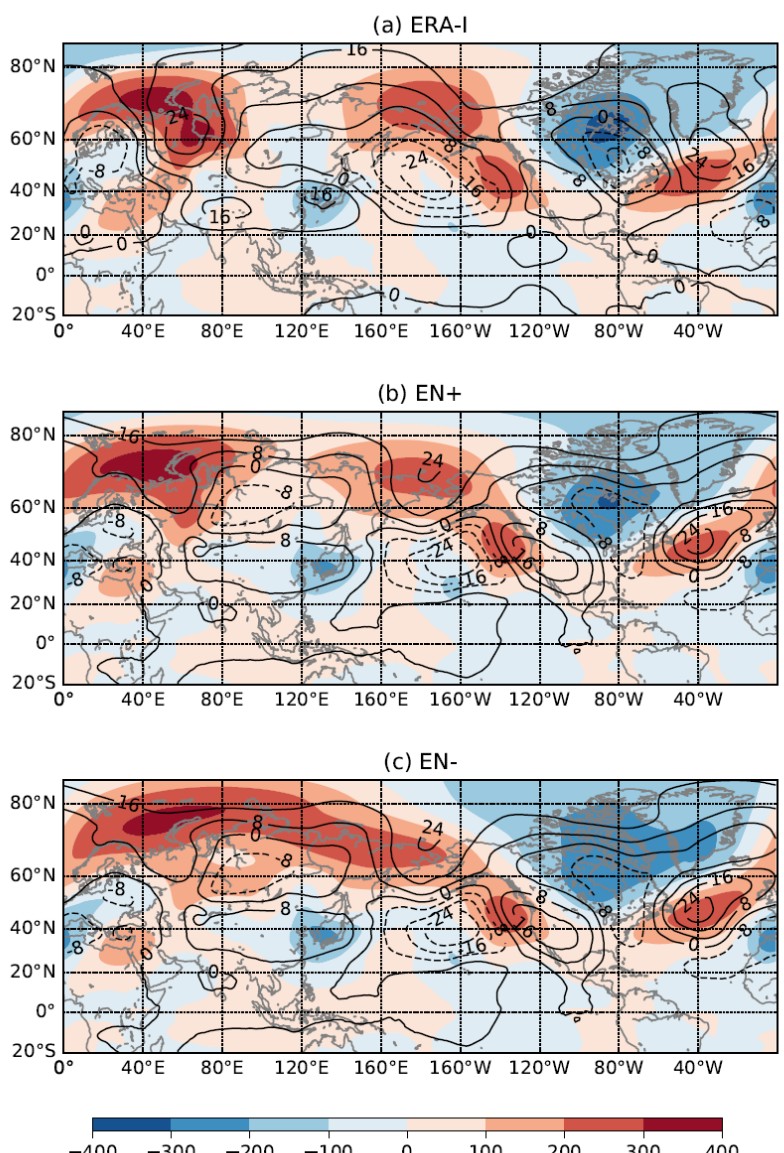

670

**Figure 14.** Composite anomalies of geopotential height (m) picked only for days with MJO phase 6 with averaged lag of 5–9 days at 500 hPa (contours) and anomalies of geopotential height at the same level averaged over 5–7 February (shaded). (a) ERA-I, composites calculated using 1980–2010 data, (b) EN+ members, composites calculated using hindcasts over 20 years (1997–2017), (c) same 675 as in (b) for EN– members.



**Appendix A**

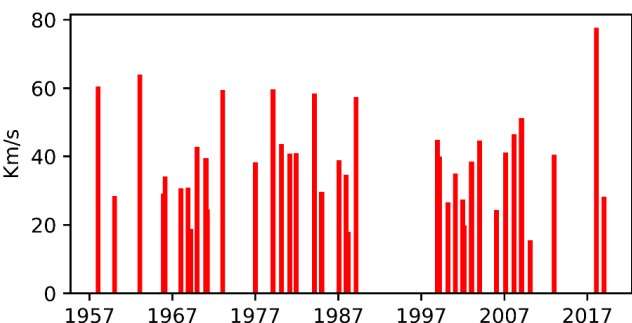

**Figure A1.** Eddy heat flux at 100 hPa (Km s$^{-1}$) averaged across 50–75° N observed over 5 days
prior to a major SSW during 1958–2018. The dates of the SSWs are taken from Charlton and Polvani
(2007) and Karpechko (2018). The heat flux in 1979–2018 was calculated using ERA-I reanalysis
while in 1958–1978 – using ERA-40 reanalysis (Uppala et al., 2005).

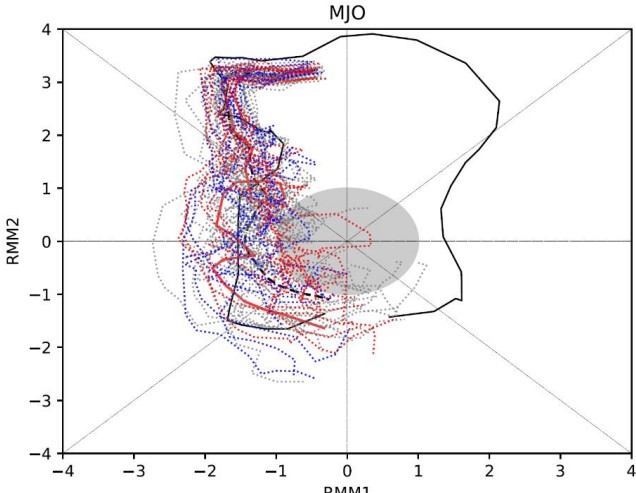

**Figure A2.** MJO phase diagram. ECMWF ensemble forecast initialized on 1 February: blue
dashed lines denote EN+ members, red dashed lines – EN– members, grey lines – all other members.
Red line denotes the control forecast, black dashed line – ensemble mean. Black solid line denotes
RMM indexes from Bureau of Meteorology (data source: http://www.bom.gov.au/climate/mjo/).