# Peer review of "Mechanisms and predictability of Sudden Stratospheric Warming in winter 2018"

_Weather and Climate Dynamics, 2020_

## Referee Comment (RC1) · Anonymous Referee #1 · 4 Jun 2020

**general comments**

This study investigates the lower tropospheric forcing of the major sudden stratospheric warming (MSSW), taking place in February 2018, and its predictability, using S2S database of the ECMWF. The main focus lies on two points: i) under which tropospheric conditions or forcings does the major warming occur (with a focus on the amplification of wave two in the stratosphere)?, and ii) why do some forecasts fail to predict the warming (within a time period of $\pm 1$ day)? Two clusters are formed emphasizing the different evolution of the polar vortex within the forecast period. MSSW 2018 occurred under an amplification of planetary wave 2 which is mainly connected with anticyclones over Alaska and the Ural Mountains. A connection is drawn to the strong MJO phase 6

roughly two weeks before the onset. The anticyclone over the Ural mountains evolves and maintains under the forcing of wave trains modified by the strong MJO phase. This scale interaction starting to diverge within the ensemble over the North Atlantic sector. It is suggested that the missing MJO response towards the Northern Hemisphere and the modification of the synoptic scale wave trains is likely the cause.

The paper is well written and clearly structured. It is the starting point to investigate which kind of processes belong to a systematic model bias and which are due to internal variability putting forward the S2S prediction system. Therefore I think the paper is worth for publishing after the authors have addressed the specific points below.

**specific comments**

*Data and Methods*

- Which resolution (spatial and temporal) is used for the ECMWF ensemble forecast?

- Were the data evaluated on 12-hourly basis or daily mean basis?

- Geopotential height anomalies are calculated with respect to different time periods for ERA-Interim (1980-2010) and forecast ensemble (hindcasts: 1997-2017). Could we expect different anomalies only due to the different time periods? Have trends been removed?

- Wave activity flux calculations can be used for different time scales spanning from e.g. 10 years down to 5-day mean averages. The investigated "quasi-stationary waves" changes under different temporal averaging. Which kind of temporal averaging is used for defining the prime quantities in the WAF calculations?

- The MJO phase space is obtained by an EOF analysis of the combined fields. Long time scales should be filtered out before the index is calculated. Is a temporal filter applied to the forecast data for calculating the MJO index based on anomalies? Is it necessary for 46-day forecasts?

*Stratospheric forecasts*

- line 141: The fluctuations of the easterlies after the MSSW onset are very interesting. Are these fluctuations a result of the vascillation cycles (Holton and Mass, 1976)? The ensemble members do not capture these vascillations? Is the tropospheric forcing maybe not steady within the ensemble in contrast to ERA-In?

- line 199-203: Figure 5 shows forecast spread at 50 hPa? Why is 50 hPa selected instead of 10 hPa (Figures before)?

- Fig. 5 and Fig.6 show ensemble spread at 50 hPa and as a cross section at 50°N, respectively. The ensemble spread is in Fig.5c above 0.3 at 50°N at 50 hPa. Why is this not visible in Fig.6c?

- In Fig. 6 the ensemble spread is gradually decreasing with height and remains below 0.1 above 150 hPa for the selected 3 dates. Why is the ensemble spread decreasing with height and remains low even if the forecast time proceeds?

*Tropospheric waves*

- Figure 9 shows the temporal evolution of the geopotential height at 250 hPa and the squared meridional wind component at the same level. Would a Hovmöller plot of the sqared meridional wind component averaged between 40°N-65°N enhance the visibility of the wave trains (like e.g. Glatt et al., 2011)? The usage of the square of anomalous meridional wind provides an alternative (e.g. Chang, 1993).

- The description of the coloured lines or coloured rectangles are missing in the captions of Figure 9 and Figure 10. A possible connection can be drawn in the paragraph starting at line 250. Is the coutour interval in Fig.10 a,b the same as in Fig.9b?

**technical corrections**

- line 94: please, remove blank after "forecast"

---

## Referee Comment (RC2) · Anonymous Referee #2 · 5 Jun 2020

General Comments: This study by Statnaia et al. examines the predictability of the February 2018 SSW by investigating the tropospheric conditions prior to the onset using the 51-member ensemble forecast of the ECMWF. In particular they focus on the role of tropospheric wave activity in the 10-14 days before the observed SSW date. They find that the Ural High region is particularly important for the onset of this SSW via the development of a blocking anticyclone in agreement with a recent study. This anticyclone was contributed to by the MJO phase 6-7, although this latter section is somewhat speculative in its nature. The paper is mostly well-written, although the English could do with some improvement (there are many places where the wrong article ['a' or 'the'] are used). Overall I find the paper quite interesting and thus warrants publication in WCD. My comments are all rather minor and hence my recommendation

is publication with minor revisions.

Specific Comments:

Lines 95-98; to clarify, you only use forecasts that are initialized on 1st February? Have you examined any forecasts initialized before (and also after) this? If so, can you say something about them? How poor was the prediction skill of such forecasts? What is the 'fraction' mentioned in the two cited papers?

Line 116; 1) 'used to localize regions on wave activity sources and sinks' → '. . .is used to identify localized regions of wave-activity sources and sinks.' 2) In this diagnostic did you calculate for stationary waves? i.e., did you average the u,v,T etc in time prior to calculating the deviations from the zonal mean? Such a calculation is important as this diagnostic is only suitable for stationary waves. For transient waves, as is more appropriate here for synoptic-scale features, the flux of Takaya and Nakamura (2001) would be more apt.

Line 162; how many ensemble members actually maintained the easterlies for the period that ERAI shows?

Line 165; are these GPH anomalies that are shown? In section 2 you mention that GPH is only shown as anomalies, although this figure does not make it clear. Further on lone 167, you mention the 4-6th February GPH fields but they are not shown in figure 3. It would be useful to include them or at least state that they are not shown if that is the case.

Lines 188-190; Such a relationship between wave1 and wave2 with one increasing and the other decreasing in amplitudes suggests some kind of wave-wave interaction occurring, i.e., wave2 grows at the expense of wave1 and vice versa. To diagnose this would be beyond the scope of this paper as it involves the enstrophy budget. However, a sentence on this may be useful as well as a suitable reference such as Smith (1983).

Line 199; What exactly is the ensemble spread shown here? The difference between

the max and min ensemble members? i.e., the best and worst ensemble members? Or between the EN+ and EN- groups?

Figure 6; The contours are too dense to be able to make out the values of the spread shown by the shading, especially in panel a. Can you decrease the number of contours in all panels?

Lines 227-228; I wonder if the Europe/Siberia sector can be considered as preconditioning the vortex prior to the reversal. Indeed, the North Atlantic sector appears to be the final straw with massive amplification just before the onset, but the Europe/Siberia sector is maximized a week or so before. Have you checked which wavenumbers dominate the flux in this figure? From figure 4 I would hazard a guess at wave-1, but it would be good to find out for sure.

Figure 9; what are the lines for? Presumably to show the blocking ridges and troughs. Please refer to them in the text and describe what they show in the caption; they could be useful in helping to explain to the reader. Further, figure 9b I find hard to understand what is going on. The features described on lines 258-265 are very difficult to see and as such I am not sure that I would agree with their characterization. For instance, the maximums in $v'^2$ on Feb 6th and 7th centered at 0E are characterized as two different events, but they could well be the same event. The max in $v'^2$ that is 80deg further downstream is characterized as part of the red-box event, but I find it rather unlikely that the feature would have travelled 80deg in one day. I find panel a much more believable and to me provides the necessary information that I would want to know; I would consider removing panel b entirely or at least making it clearer in the text exactly how you are tracking the features.

Figure 11; Can you include the u contours on this plot to show how the winds and wave propagation are related. Further, the panels of the composite EN+ and EN- would be helpful to see how overall, the best and worst ensemble members capture the horizontal wave propagation compared to ERAI.

Figure A2; over how many days are these trajectories run?

Line 318; Why is the central date here cast as February 7th? The central date in ERAI was 12th February throughout the earlier manuscript. Is this chosen as the period before which the vortex started being displaced and then splitting?

Line 332 and 339; how sensitive are the results in figure 14 to different lag stages? Given the previous comment, this can be important. Why exactly are lags 5-9 days before February 7th chosen? Figure 14; it is somewhat unclear from the caption and text exactly what this figure is showing. The contours are the 'climatological' response to every MJO phase 6-7 event, delayed by 5-9 days, in the ∼20 years of data? Then the shading represents the GPH anomalies just for the period 5-7th February for this one SSW event? Hence if the shading projects onto the contours then one could say there is constructive interference? Please clarify.

Line 361; I would say that the EN- composite actually captures the global pattern pretty well in ERAI and EN+. It is the difference over the Urals/East Asia that is most pronounced as the observed wave-2 pattern is instead a wave-1 via a connection of the ridges.

Summary; Please include references to figure numbers throughout.

Technical Comments:

Line 40; add 'a' before 'negative phase of the...'

Line 101; what does EN stand for here? It is a little confusing as it can be easily mixed up with the El Nino phase (indeed initially I thought that was what it meant until I got to figure 2). Can you use a more appropriate acronym?

line 225; 'third' → 'sector'

Line 255; 'cost' → 'coast'

Line 397; The sentence starting 'Here we also' does not make sense written as it.

[Figure]

Please change to make clear.

Lines 413-414; Rewrite sentence as 'the composite analysis provides evidence, albeit indecisive, that teleconnections...'

References: Smith 1983, Observations of wave-wave interactions in the stratosphere. JAS.

---

## Referee Comment (RC3) · Anonymous Referee #3 · 10 Jun 2020

The present manuscript analyses the sudden stratospheric warming that took place in mid-February 2018 (SSW2018). In particular, the study focuses on the tropospheric forcing of this phenomenon by examining its predictability based on the ECMWF ensemble forecast of the S2S initiative. The SSW2018 is found to be preceded by an amplification of wavenumber 2 wave activity in the stratosphere that is linked to the occurrence of a blocking in the Ural Mountains region. The authors also investigate the role of the record-breaking Madden Julian Oscillation (MJO) phase 6 in triggering the SSW event. The results show that this phenomenon might help, although its influence does not seem to be decisive.

The manuscript is well-written and the analysis is interesting. Thus, my recommendation is publication after having performed some minor changes.

[Figure]

Specific comments:

L42-43: I think the clearest example of the interdecadal variability of SSW is the 2000s decade when there was an SSW in almost every winter and the 1990s decade with a very low frequency of SSWs.

L57: Please note that some studies such as de la Cámara et al (2019) have also shown that it is not always necessary to have an enhancement of tropospheric waves for the occurrence of an SSW.

L70: This was also shown by Ayarzagüena et al. (2018).

L110-112: Is the data detrended?

L115-123: Instead of the wave activity flux by Plumb (1985), I would suggest using the wave activity flux by Takaya and Nakamura (2001). This flux is defined for the case of a zonally varying basic flow, which, I think, is more appropriate in this study. The basic state in the Northern Hemisphere in winter shows inhomogeneities that can modulate the propagation of Rossby wave packets. Takaya and Nakamura's flux only focuses on the wave activity associated with Rossby wave packets, as the wavy anomalies are considered to be embedded in the basic flow that includes the climatological planetary waves. Actually, this flux was used by different authors to study tropospheric forcing of SSWs such as the event of January 2006 (Nishii et al. 2009) or the SSWs of 2009 and 2010 (Ayarzagüena et al., 2011).

L147-157: This evolution of the polar night jet (PNJ) is typical of split-vortex SSWs (S SSWs) (Charlton and Polvani, 2007). Before these events, the PNJ typically shifts poleward and then, the vortex splits into two pieces. Albers and Birner (2014) also show that the polar vortex before S SSWs tends to be constrained around the pole and has little vertical tilt. I think some comment about that could be added.

L205-207 and figure 6: it is difficult for me to identify the regions with large ensemble-forecast spread.

[Figure]

L217-229: I agree with the authors that there are some bursts of wave activity in the troposphere before the occurrence of SSW2018. I also agree on the enhancement of wave activity in the stratosphere, particularly in the North Atlantic sector. However, I have the impression that apart from the tropospheric forcing there is a self-amplification of the wave activity in the stratosphere. These results would be also consistent with the characteristics of wave activity during S SSWs highlighted by previous studies. For instance, Plumb (1981) and Albers and Birner (2014) indicate that it is typical for S SSWs that an initial vortex structure close to its resonant point can split the vortex with only a small increase in tropospheric wave forcing. I would suggest adding some comments about that in the text.

L241-245: When split into three regions, the correlation coefficient between the vertical component of the WAF forecasts on 4–11 February and U10 forecasts on 12 February is not statistically significant in the troposphere. Do you know why?

L257-265: I must confess I find it difficult to see the propagation of synoptic structures in Figure 9b. In this sense, I am not 100% sure that the anomalies of v250ˆ2 on 8 February around 80°E are related to the anomalies over the Eastern Atlantic at the beginning of February, as the red box in figure 9b seems to indicate. There are already some anomalies at high latitudes in Eurasia on 6 February that seem to intensify in the following days. A similar evolution is detected in Figure 10a for EN+ members, but in EN- members you have a very similar pattern over the North Atlantic on 3-6 February, but the development of the anomalies over the Eurasia is missing.

L321: please add "the"

L397: we have also shown

L397-399: please rewrite this sentence.

L419: Domeisen et al. 2019a or b?

Figure 14, caption: It is not 100% clear for me what you are showing in contours. Is it

the geopotential height anomalies for all MJO phase 6 in the whole period of study? I understood so, but it would be great if you indicate it more clearly in the text.

---

## Author Comment (AC1) · 20 Jul 2020

**Response to Reviewers**

We would like to thank the three reviewers for careful reading, insightful and constructive comments and helpful suggestions for our study. All comments have been addressed in the revised manuscript. We answer point by point to the comments and suggestions below. Reviewers' comments are included below in black font colour and our replies in blue. The revised and clarifying figures are included at the end of this document.

**Reviewer 1:**

**General comments**

This study investigates the lower tropospheric forcing of the major sudden stratospheric warming (MSSW), taking place in February 2018, and its predictability, using S2S database of the ECMWF. The main focus lies on two points: i) under which tropospheric conditions or forcings does the major warming occur (with a focus on the amplification of wave two in the stratosphere)?, and ii) why do some forecasts fail to predict the warming (within a time period of  $\pm 1$  day)? Two clusters are formed emphasizing the different evolution of the polar vortex within the forecast period. MSSW 2018 occurred under an amplification of planetary wave 2 which is mainly connected with anticyclones over Alaska and the Ural Mountains. A connection is drawn to the strong MJO phase 6 roughly two weeks before the onset. The anticyclone over the Ural mountains evolves and maintains under the forcing of wave trains modified by the strong MJO phase. This scale interaction starting to diverge within the ensemble over the North Atlantic sector. It is suggested that the missing MJO response towards the Northern Hemisphere and the modification of the synoptic scale wave trains is likely the cause. The paper is well written and clearly structured. It is the starting point to investigate which kind of processes belong to a systematic model bias and which are due to internal variability putting forward the S2S prediction system. Therefore I think the paper is worth for publishing after the authors have addressed the specific points below.

**Specific comments**

Data and Methods

• Which resolution (spatial and temporal) is used for the ECMWF ensemble forecast?

The horizontal resolution used for the ECMWF ensemble forecast is  $1^{\circ} \times 1^{\circ}$  and the temporal resolution is 12 hours. We added information about the resolution to the Data and Methods section.

• Were the data evaluated on 12-hourly basis or daily mean basis?

The data were evaluated on the 12-hourly basis. We added this to the Data and Methods section.

• Geopotential height anomalies are calculated with respect to different time periods for ERA-Interim (1980-2010) and forecast ensemble (hindcasts: 1997-2017). Could we expect different anomalies only due to the different time periods? Have trends been removed?

We used the period of 1980-2010 as the longer the period is useful for cancelling the noise. However, in order to address this comment, we calculated ERA-Interim geopotential height anomalies with respect for the same time period as for the forecast ensemble: 1997-2017. We also removed trends in both datasets. The results show that the anomalies fields do not differ qualitatively due to the different time periods and detrending and our choices do not influence our conclusions (Figure AC1).

• Wave activity flux calculations can be used for different time scales spanning from e.g. 10 years down to 5-day mean averages. The investigated "quasi-stationary waves" changes under different temporal averaging. Which kind of temporal averaging is used for defining the prime quantities in the WAF calculations?

Wave activity flux (WAF) vectors can be indeed applied for different time scales, but due to the fast development of the SSW2018 we are using 3-day averages in the calculation of the Plumb flux. The Plumb (1985) formulation of the wave activity flux was used to study the SSWs occurred in 2009 (Harada et al., 2010) and in 2013 (Coy and Pawson, 2015) to identify the source regions and propagation of the wave activity.

Another formulation of WAF by Takaya and Nakamura (2001) given for zonally varying basic flow which can be used for illustrating an instantaneous status of the three dimensional wave propagation gives similar results to the Plumb WAF (Figure AC2). Both fluxes fields show good agreement in our case.

• The MJO phase space is obtained by an EOF analysis of the combined fields. Long time scales should be filtered out before the index is calculated. Is a temporal filter applied to the

**forecast data for calculating the MJO index based on anomalies? Is it necessary for 46-day forecasts?**

The indexes were calculated at ECMWF using the Wheeler and Hendon (2004) method. According to this method the influence of seasonal cycle and interannual variability is removed before calculating the EOFs. The RMM indexes forecasted by the ECMWF extended-range prediction system can be obtained from the Subseasonal-to-Seasonal Prediction Project: http://s2sprediction.net/. This information is added to manuscript.

**Stratospheric forecasts**

• line 141: The fluctuations of the easterlies after the MSSW onset are very interesting. Are these fluctuations a result of the vascillation cycles (Holton and Mass, 1976)? The ensemble members do not capture these vascillations? Is the tropospheric forcing maybe not steady within the ensemble in contrast to ERA-In?

According to Holton and Mass (1976) large-scale stratospheric motions may vacillate in the irregular manner even with steady tropospheric forcing. However, the amplitude of this forcing plays the key role and defines if the oscillations in stratosphere are damped or not. Both period and amplitude of the stratospheric oscillations depend greatly on the strength of the initial forcing. In the SSW2018 case the oscillation period is approximately 4-6 days in ERA-I. The nature of these fluctuations is not clear to us, however we do not think that they are related to the Holton and Mass (1976) oscillations as the observed oscillations are too short to be associated with the Holton and Mass (1976) vacillations. Figure 2 from Holton and Mass (1976) suggests approximately 40-60 days cycle, depending on the amplitude of the tropospheric forcing. Period of these oscillations should be comparable to the period of radiation recovery in the lower stratosphere which controls the wave propagation, ca. 20 days. The fluctuations of zonal mean zonal wind in February 2018 might be induced by the horizontal propagation of waves and is likely connected to the vortex movement. Although the analysis of the vacillations of the easterlies after the SSW2018 onset in the re-analysis and forecast data seems to be an interesting topic it lies beyond the scope of our present paper.

• line 199-203: Figure 5 shows forecast spread at 50 hPa? Why is 50 hPa selected instead of 10 hPa (Figures before)?

We thank the reviewer for pointing that out and have changed the level in the Figure 5 to 10 hPa for consistency reasons. (Figure AC3). The geographical distribution of spread remains similar on both levels.

• Fig. 5 and Fig.6 show ensemble spread at 50 hPa and as a cross section at 50°N, respectively. The ensemble spread is in Fig.5c above 0.3 at 50°N at 50 hPa. Why is this not visible in Fig.6c?

In Figure 6 of the manuscript the spread and anomaly are normalized by pressure, i.e. are multiplied by (p/1000 hPa) and its square root respectively, for display reasons to facilitate the comparison across the different pressure levels. In Figure AC4 we show the spread that is not normalized by pressure. We believe that there is no disagreement with Fig. 5c of the manuscript.

• In Fig. 6 the ensemble spread is gradually decreasing with height and remains below 0.1 above 150 hPa for the selected 3 dates. Why is the ensemble spread decreasing with height and remains low even if the forecast time proceeds?

In Figure 6 spread is normalized by pressure for display reasons. Here we include the figure for non-normalized spread (Figure AC4). The spread is also normalized by the minimum and maximum values within the domain north of 20° N for each day separately to highlight the geographical spread rather than its growth with time.

**Tropospheric waves**

• Figure 9 shows the temporal evolution of the geopotential height at 250 hPa and the squared meridional wind component at the same level. Would a Hovmöller plot of the sqared meridional wind component averaged between 40°N-65°N enhance the visibility of the wave trains (like e.g. Glatt et al., 2011)? The usage of the square of anomalous meridional wind provides an alternative (e.g. Chang, 1993).

In order to account for this comment, we plot a Hovmöller diagram of the squared meridional wind averaged over 40°N-65°N (Figure AC5). However, we do not find it more clear for tracking the wave trains in our case. In the manuscript we use the Nishii and Nakamura (2010) approach for highlighting the wave trains.

• The description of the coloured lines or coloured rectangles are missing in the captions of Figure 9 and Figure 10. A possible connection can be drawn in the paragraph starting at line 250. Is the coutour interval in Fig.10 a,b the same as in Fig.9b?

The description of the colored lines is added to the captions (Figure AC6 and AC7) and mentioned in text as suggested. The contour intervals in Figures AC6b and AC7 a, b (Figures 9b and 10 a,b in manuscript, respectively) are the same:  $800 \text{ m}^2 \text{ s}^{-2}$ .

**technical corrections**

• line 94: please, remove blank after "forecast"

Removed

**Reviewer 2:**

**General comments**

This study by Statnaia et al. examines the predictability of the February 2018 SSW by investigating the tropospheric conditions prior to the onset using the 51-member ensemble forecast of the ECMWF. In particular they focus on the role of tropospheric wave activity in the 10-14 days before the observed SSW date. They find that the Ural High region is particularly important for the onset of this SSW via the development of a blocking anticyclone in agreement with a recent study. This anticyclone was contributed to by the MJO phase 6-7, although this latter section is somewhat speculative in its nature. The paper is mostly well-written, although the English could do with some improvement (there are many places where the wrong article ['a' or 'the'] are used). Overall I find the paper quite interesting and thus warrants publication in WCD. My comments are all rather minor and hence my recommendation is publication with minor revisions.

**Specific comments**

Lines 95-98; to clarify, you only use forecasts that are initialized on 1st February? Have you examined any forecasts initialized before (and also after) this? If so, can you say something about them? How poor was the prediction skill of such forecasts? What is the 'fraction' mentioned in the two cited papers?

We thank the reviewer for this question. We focus in this study only on the forecast initialized on 1 February, because it is the first date when the forecast substantially changes from not showing the mean zonal wind reversal (Lee et al., 2019) to 14 ensemble members out of 51 predicting the reversal within 1 day from the observed onset date (Figure 1 of our manuscript). The ECMWF ensemble forecast is produced only twice a week (Monday and Thursday) and according to the forecast initialized on the 29 January no members showed reversal to easterlies (Lee et al., 2019). But the odds of an SSW in the forecast initialized on the 1 February increased by 2.5 times compared to the climatological frequency of easterlies on any February day which is 0.11 (Karpechko et al., 2018). On the other hand, with the reduction of the lead time the prediction skill increases rapidly and on 5 February the ensemble mean predict the negative zonal mean zonal wind (Karpechko et al., 2018). In order to investigate the driving mechanisms of the SSW2018 but not predictability as a function of lead time we focus on the first forecast showing the abrupt change to predicting the possibility of the event. Line 116; 1) 'used to localize regions on wave activity sources and sinks' -> '...is used to identify localized regions of wave-activity sources and sinks.' 2) In this diagnostic did you calculate for stationary waves? i.e., did you average the u,v,T etc in time prior to calculating the deviations from the zonal mean? Such a calculation is important as this diagnostic is only suitable for stationary waves. For transient waves, as is more appropriate here for synoptic-scale features, the flux of Takaya and Nakamura (2001) would be more apt.

1) Thank you, we corrected the sentence.

2) Thank you for pointing this out, we averaged all the parameters as 3-day mean and used the 5-7 February fields for analysis. But as the other reviewers also pointed out the more appropriate use of the Takaya and Nakamura (2001) flux, we show it here as well (Figure AC2). Both fluxes fields show good agreement in our case.

Line 162; how many ensemble members actually maintained the easterlies for the period that ERAI shows?

Only 2 ensemble members maintained the easterlies until the end of February and returned to westerlies within 2 days from the observed reversal.

Line 165; are these GPH anomalies that are shown? In section 2 you mention that GPH is only shown as anomalies, although this figure does not make it clear. Further on lone 167, you mention the 4-6th February GPH fields but they are not shown in figure 3. It would be useful to include them or at least state that they are not shown if that is the case.

We thank the reviewer for pointing this out. In Figure 3 the full geopotential fields are shown and we corrected this in Section 2 Data and Methods. We do not show the 4-6 February fields for space sake and we now mention this in text.

Lines 188-190; Such a relationship between wave1 and wave2 with one increasing and the other decreasing in amplitudes suggests some kind of wave-wave interaction occurring, i.e., wave2 grows at the expense of wave1 and vice versa. To diagnose this would be beyond the scope of this paper as it involves the enstrophy budget. However, a sentence on this may be useful as well as a suitable reference such as Smith (1983).

We agree with the reviewer about this point and therefore we have added the following sentence to the manuscript: "Moreover, the amplitude vacillation between PW1 and PW2 may be caused by wave-wave interactions because enstrophy and energy must be conserved (Smith, 1983)."

Line 199; What exactly is the ensemble spread shown here? The difference between the max and min ensemble members? i.e., the best and worst ensemble members? Or between the EN+ and EN- groups?

In Figures 5 and 6 of the manuscript we show the ensemble spread in geopotential height which is a measure of the difference between the members and is represented by the standard deviation with respect to the ensemble mean:

$$Spread = \frac{\sqrt{\frac{1}{N}\sum_{i=1}^{N}(g_i - \bar{g})}}{\bar{g}} \tag{1}$$

where  $g_i$  is geopotential height of an ensemble member,  $\bar{g}$  – ensemble mean, N – number of ensemble members (N=51).

We use the spread to assess the uncertainty in the forecast as small spread indicates high theoretical forecast accuracy, while large spread indicates low theoretical forecast accuracy.

Figure 6; The contours are too dense to be able to make out the values of the spread shown by the shading, especially in panel a. Can you decrease the number of contours in all panels? We have decreased the number of contours for a better visibility (Figure AC4).

Lines 227-228; I wonder if the Europe/Siberia sector can be considered as preconditioning the vortex prior to the reversal. Indeed, the North Atlantic sector appears to be the final straw with massive amplification just before the onset, but the Europe/Siberia sector is maximized a week or so before. Have you checked which wavenumbers dominate the flux in this figure? From figure 4 I would hazard a guess at wave-1, but it would be good to find out for sure.

We agree that the Siberian sector could have contributed to the preconditioning of the stratosphere during the period of wavenumber 1 amplification. We will revise the manuscript to address this point.

Figure 9; what are the lines for? Presumably to show the blocking ridges and troughs. Please refer to them in the text and describe what they show in the caption; they could be useful in helping to explain to the reader. Further, figure 9b I find hard to understand what is going on. The features described on lines 258-265 are very difficult to see and as such I am not sure that I would agree with their characterization. For instance, the maximums in v'^2 on Feb 6th and 7th centered at 0E are characterized as two different events, but they could well be the same event. The max in v'^2 that is 80deg further downstream is characterized as part of the red-box

event, but I find it rather unlikely that the feature would have travelled 80deg in one day. I find panel a much more believable and to me provides the necessary information that I would want to know; I would consider removing panel b entirely or at least making it clearer in the text exactly how you are tracking the features.

We have reworked the figures to make them more clear (Figures AC6 and AC7). The coloured lines suggest the propagation of wave trains; in the revised manuscript we have added this to the figures captions and in text. We do not suggest that the anomalies of v2502 on 8 February around 80°E are part of the same wave train, but rather of the blocking which had appeared on 6 February at high latitudes.

Figure 11; Can you include the u contours on this plot to show how the winds and wave propagation are related. Further, the panels of the composite EN+ and EN- would be helpful to see how overall, the best and worst ensemble members capture the horizontal wave propagation compared to ERAI.

Please see the ERA-I zonal wind contours in Figure AC8 below. We find that adding ucontours to the manuscript figure makes it too noisy and is not very informative.

We find the separate panels of the composite EN+ and EN- yield less information than the difference between them, because the differences between the two clusters are practically indiscernible in this case (Figure AC9).

Figure A2; over how many days are these trajectories run? The length of the ECMWF extended-range ensemble forecast is 46 days.

Line 318; Why is the central date here cast as February 7th? The central date in ERAI was 12th February throughout the earlier manuscript. Is this chosen as the period before which the vortex started being displaced and then splitting?

We thank the reviewer for this correction and have changed the text in the following way: "Before the SSW2018 central date an active MJO in phases 6 and 7 with large amplitude prevailed in tropical Indian ocean and South China Sea" as the MJO in phases 6 and 7 took place from 27 January until 18 February.

Line 332 and 339; how sensitive are the results in figure 14 to different lag stages? Given the previous comment, this can be important. Why exactly are lags 5-9 days before February 7th chosen? Figure 14; it is somewhat unclear from the caption and text exactly what this figure is

showing. The contours are the 'climatological' response to every MJO phase 6-7 event, delayed by 5-9 days, in the \_20 years of data? Then the shading represents the GPH anomalies just for the period 5-7th February for this one SSW event? Hence if the shading projects onto the contours then one could say there is constructive interference? Please clarify.

The MJO phase 6 took place on 27–31 January, therefore the effect of this tropical phenomena should be studied in the extratropics with a time lag. The lag of 5–9 days used in our study roughly corresponds to the period in the early February when the tropospheric waves forced SSW2018 (Figure 9 in the manuscript). The results are quite robust to different lags however the lag of 5–9 days yield the clearest fingerprint. In Figure 14 the contours indeed show the climatological (1980–2010) response to MJO phase 6 with the chosen lag superimposed on the anomalies of geopotential height for 5–7 February 2018 (shading) for comparison. The juxtaposition of these two fields on the same plot highlights that, apart from some dissimilarity in the positions and strength of the features, their overall structure is well captured by the model. The geopotential height anomalies for 5–7 February 2018 are calculated using hindcasts for the previous 20 years produced on-the-fly with the real-time forecasts (1997-2010) which represent the model's own climatology. For consistency we also recalculated the ERA-I climatology using the same period as for the re-forecasts however this does not change the results significantly (Figure AC1). We have updated the text to make it clearer by specifying what features we are pointing out.

Line 361; I would say that the EN- composite actually captures the global pattern pretty well in ERAI and EN+. It is the difference over the Urals/East Asia that is most pronounced as the observed wave-2 pattern is instead a wave-1 via a connection of the ridges.

Indeed, the difference over the Urals with two anticyclones merged is the most pronounced and it seems to us to be crucial in the SSW2018 case. To highlight this we have rewritten the text in the following way: "On the contrary, the response in the EN– cluster shows a PW1 pattern with two highs in Alaska and Ural region merged together (Fig. 14c), which is consistent with the EN– forecasts not capturing the amplification of PW2 in the stratosphere (Fig. 4c)."

Summary; Please include references to figure numbers throughout.

Reference to figure numbers is included in text. Thank you.

**Technical comments**

Line 40; add 'a' before 'negative phase of the...' Added Line 101; what does EN stand for here? It is a little confusing as it can be easily mixed up with the El Nino phase (indeed initially I thought that was what it meant until I got to figure 2). Can you use a more appropriate acronym?

We changed EN to ENS, which stands for 'ensemble'. Thank you.

line 225; 'third' -> 'sector'

**Corrected**

Line 255; 'cost' -> 'coast'

Corrected

Line 397; The sentence starting 'Here we also' does not make sense written as it. Please change to make clear.

We have rewritten the sentence in the following way: "We also showed that the wave packet crucial for the Ural blocking is not captured by the ensemble members that failed to forecast SSW2018."

Lines 413-414; Rewrite sentence as 'the composite analysis provides evidence, albeit indecisive, that teleconnections...'

Rewritten as proposed, thank you.

References: Smith 1983, Observations of wave-wave interactions in the stratosphere. JAS.

**Reviewer 3:**

**General comments**

The present manuscript analyses the sudden stratospheric warming that took place in mid-February 2018 (SSW2018). In particular, the study focuses on the tropospheric forcing of this phenomenon by examining its predictability based on the ECMWF ensemble forecast of the S2S initiative. The SSW2018 is found to be preceded by an amplification of wavenumber 2 wave activity in the stratosphere that is linked to the occurrence of a blocking in the Ural Mountains region. The authors also investigate the role of the record-breaking Madden Julian Oscillation (MJO) phase 6 in triggering the SSW event. The results show that this phenomenon might help, although its influence does not seem to be decisive.

The manuscript is well-written and the analysis is interesting. Thus, my recommendation is publication after having performed some minor changes.

**Specific comments**

L42-43: I think the clearest example of the interdecadal variability of SSW is the 2000s decade when there was an SSW in almost every winter and the 1990s decade with a very low frequency of SSWs.

We thank the reviewer for pointing that out. We have updated the text in the following way: "SSWs occur approximately once every second winter; however, there is no regularity: during the 1990s decade SSWs occurred only twice while in the 2000s they took place almost every winter and during the last decade the events occurred in 2013, 2018, 2019."

L57: Please note that some studies such as de la Cámara et al (2019) have also shown that it is not always necessary to have an enhancement of tropospheric waves for the occurrence of an SSW.

We thank the reviewer for this comment and have added it to the text: "However, SSWs are not always preceded by anomalous tropospheric wave activity. Some recent studies point out that the lower stratosphere dynamics and vortex geometry play role in SSW onset (De La Cámara et al., 2019)."

L70: This was also shown by Ayarzagüena et al. (2018). We added the reference to Ayarzagüena et al., (2018) in text.

**L110-112: Is the data detrended?**

As detrending was also pointed out by another reviewer we removed trends in datasets. Results show that the detrended fields do not differ qualitatively (e.g. see Figure AC1).

L115-123: Instead of the wave activity flux by Plumb (1985), I would suggest using the wave activity flux by Takaya and Nakamura (2001). This flux is defined for the case of a zonally varying basic flow, which, I think, is more appropriate in this study. The basic state in the Northern Hemisphere in winter shows inhomogeneities that can modulate the propagation of Rossby wave packets. Takaya and Nakamura's flux only focuses on the wave activity associated with Rossby wave packets, as the wavy anomalies are considered to be embedded in the basic flow that includes the climatological planetary waves. Actually, this flux was used by different authors to study tropospheric forcing of SSWs such as the event of January 2006 (Nishii et al. 2009) or the SSWs of 2009 and 2010 (Ayarzagüena et al., 2011).

Thank you for pointing this out. As the other reviewers have also pointed out the more appropriate use of the Takaya and Nakamura (2001) flux, we show it here as well (Figure AC2). Both fluxes fields show good agreement in our case. For calculating the flux by Plumb (1985) we averaged all the parameters as 3-day mean to account for the stationary waves.

L147-157: This evolution of the polar night jet (PNJ) is typical of split-vortex SSWs (S SSWs) (Charlton and Polvani, 2007). Before these events, the PNJ typically shifts poleward and then, the vortex splits into two pieces. Albers and Birner (2014) also show that the polar vortex before S SSWs tends to be constrained around the pole and has little vertical tilt. I think some comment about that could be added.

We have added the following in the text: "The unique prewarming vortex evolution before split SSWs characterized by funnel-shaped vortex geometry and little vertical tilt that is pointed out by Albers and Birner (2014)."

L205-207 and figure 6: it is difficult for me to identify the regions with large ensemble forecast spread.

In Figure 6 spread and anomaly are normalized by pressure, i.e. are multiplied by (p/1000 hPa) and its square root, respectively for display reasons. In Figure AC4 we show the spread that is not normalized by pressure.

L217-229: I agree with the authors that there are some bursts of wave activity in the troposphere before the occurrence of SSW2018. I also agree on the enhancement of wave activity in the stratosphere, particularly in the North Atlantic sector. However, I have the impression that apart from the tropospheric forcing there is a self-amplification of the wave activity in the stratosphere. These results would be also consistent with the characteristics of wave activity during S SSWs highlighted by previous studies. For instance, Plumb (1981) and Albers and Birner (2014) indicate that it is typical for S SSWs that an initial vortex structure close to its resonant point can split the vortex with only a small increase in tropospheric wave forcing. I would suggest adding some comments about that in the text.

We added these results in the text: "The anomalous planetary wave forcing from the troposphere might alter the geometry of the vortex hence precondition it to splitting by triggering the internal resonance (Albers and Birner, 2014)."

L241-245: When split into three regions, the correlation coefficient between the vertical component of the WAF forecasts on 4–11 February and U10 forecasts on 12 February is not statistically significant in the troposphere. Do you know why?

The reviewer raises a very interesting question, but we can only speculate about why the correlation coefficient is not significant in the troposphere which might lead to more confusion than clarity.

L257-265: I must confess I find it difficult to see the propagation of synoptic structures in Figure 9b. In this sense, I am not 100% sure that the anomalies of v2502 on 8 February around 80°E are related to the anomalies over the Eastern Atlantic at the beginning of February, as the red box in figure 9b seems to indicate. There are already some anomalies at high latitudes in Eurasia on 6 February that seem to intensify in the following days. A similar evolution is detected in Figure 10a for EN+ members, but in EN- members you have a very similar pattern over the North Atlantic on 3-6 February, but the development of the anomalies over the Eurasia is missing.

We have reworked the figures to make them more clear (Figures AC6 and AC7). The coloured lines suggest propagation of wave trains, in the revised manuscript we have added this to the figures' captions and in text. We do not suggest that the anomalies of v2502 on 8 February around 80°E are part of the same wave train, we agree with the reviewer that they are part of the blocking which had appeared on 6 February at high latitudes. We point out the missing

development of the anomalies in EN- members in text as one of the most prominent differences between the two ensemble members groups.

L321: please add "the"

**Added**

L397: we have also shown

We have rewritten this sentence in the following way: "We also showed that the wave packet crucial for the Ural blocking is not captured by the ensemble members that failed to forecast SSW2018."

L397-399: please rewrite this sentence.

We have rewritten this sentence in the following way: "We also showed that the wave packet crucial for the Ural blocking is not captured by the ensemble members that failed to forecast SSW2018."

L419: Domeisen et al. 2019a or b?

Corrected: Domeisen et al., 2019a

Figure 14, caption: It is not 100% clear for me what you are showing in contours. Is it the geopotential height anomalies for all MJO phase 6 in the whole period of study? I understood so, but it would be great if you indicate it more clearly in the text.

We clarified this in text: "In Fig. 14a contours represent the composite fields showing the climatological ERA-I fingerprint of the MJO phase 6. Contours in Figures 14 b and c show similar fingerprint but constructed with the model hindcasts over the 20-years period."

**References:**

Albers, J. R. and Birner, T.: Vortex Preconditioning due to Planetary and Gravity Waves prior to Sudden Stratospheric Warmings, J. Atmos. Sci., 71(11), 4028–4054, doi:10.1175/jas-d-14-0026.1, 2014.

Ayarzagüena, B., Barriopedro, D., Garrido-Perez, J. M., Abalos, M., de la Cámara, A., García-Herrera, R., Calvo, N. and Ordóñez, C.: Stratospheric Connection to the Abrupt End of the 2016/2017 Iberian Drought, Geophys. Res. Lett., 45(22), 12,639-12,646, doi:10.1029/2018GL079802, 2018.

Coy, L. and Pawson, S.: The major stratospheric sudden warming of January 2013: Analyses and forecasts in the GEOS-5 data assimilation system, Mon. Weather Rev., 143(2), 491–510, doi:10.1175/MWR-D-14-00023.1, 2015.

Harada, Y., Goto, A., Hasegawa, H., Fujikawa, N., Naoe, H. and Hirooka, T.: A Major Stratospheric Sudden Warming Event in January 2009, J. Atmos. Sci., 67(6), 2052–2069, doi:10.1175/2009jas3320.1, 2010.

Holton, J. R. and Mass, C.: Stratospheric vascillation cycles, J. Atmos. Sci., 33, 2218–2225 [online] Available from: https://doi.org/10.1175/1520-0469(1976)033%3C2218:SVC%3E2.0.CO;2%0A, 1976.

Karpechko, A. Y., Charlton-Perez, A., Balmaseda, M., Tyrrell, N. and Vitart, F.: Predicting Sudden Stratospheric Warming 2018 and Its Climate Impacts With a Multimodel Ensemble, Geophys. Res. Lett., 45(24), 13,538-13,546, doi:10.1029/2018GL081091, 2018.

De La Cámara, A., Birner, T. and Albers, J. R.: Are sudden stratospheric warmings preceded by anomalous tropospheric wave activity?, J. Clim., 32(21), 7173–7189, doi:10.1175/JCLI-D-19-0269.1, 2019.

Lee, S. H., Charlton-Perez, A. J., Furtado, J. C. and Woolnough, S. J.: Abrupt stratospheric vortex weakening associated with North Atlantic anticyclonic wave breaking, J. Geophys. Res. Atmos., 2019JD030940, doi:10.1029/2019JD030940, 2019.

Nishii, K. and Nakamura, H.: Three-dimensional evolution of ensemble forecast spread during the onset of a stratospheric sudden warming event in January 2006, Q. J. R. Meteorol. Soc., 136(649), 894–905, doi:10.1002/qj.607, 2010.

Plumb, R. A.: On the Three-Dimensional Propagation of Stationary Waves, J. Atmos. Sci.,

42(3), 217–229, doi:10.1175/1520-0469(1985)042<0217:ottdpo>2.0.co;2, 1985.

Smith, A. K.: Observation of Wave-Wave Interactions in the Stratosphere, J. Atmos. Sci., 40, 2484–2496, doi:https://doi.org/10.1175/1520-0469(1983)040<2484:OOWWII>2.0.CO;2, 1983.

Takaya, K. and Nakamura, H.: A Formulation of a Phase-Independent Wave-Activity Flux for Stationary and Migratory Quasigeostrophic Eddies on a Zonally Varying Basic Flow, J. Atmos. Sci., 58(6), 608–627, doi:10.1175/1520-0469(2001)058<0608:afoapi>2.0.co;2, 2001.

Wheeler, M. C. and Hendon, H. H.: An All-Season Real-Time Multivariate MJO Index : Development of an Index for Monitoring and Prediction, Mon. Weather Rev., 132, 1917–1932, 2004.

---

## Referee Report (RR1)

**Review: Mechanisms and predictability of Sudden stratospheric warming in winter 2018**

**by Irina A. Statnaia, Alexey Yu. Karpechko, Heikki J. Järvinen**

Thanks for the revision. Except for two small points that could still be changed but it is not necessary, the article is worth being published in its current state.

1. Why is Figure 8 not changing if WAF according to Takaya and Nakamura is used?

2. Figure 14: For ERA-I, the wave trains associated with MJO phase 6 (contour lines) over the North Pacific and North America, and North Atlantic (Figure AC1) can be seen more clearly in Figure AC1 ( including detrending and same period) in comparison to Fig. 14. Furthermore, the anomalies (shaded areas) show a better agreement, especially over North America and the North Atlantic. I suggest to include Figure AC1 instead of Figure 14.

---

## Author Response (AR2)

**Response to Reviewers**

We thank the reviewers for their helpful comments.

**Reviewer 1:**

1. Why is Figure 8 not changing if WAF according to Takaya and Nakamura is used? We thank the reviewer for noting this and update the figure using the Takaya and Nakamura flux. We also updated the text (lines 272-275): "The correlation analysis of the zonal mean WAF at each level averaged over 4–11 February with forecast U10 on 12 February across individual ensemble members shows the negative correlation, starting from the upper troposphere, at 0.05 significance level (Fig. 8a)."

Similarly, we also update Figures 7 and 13 of the manuscript using the Takaya and Nakamura flux. New Figure 13 suggested importance of wave activity propagation over North Atlantic, which was not so clear when using Plumb flux. Following this change, we updated the text as follows (lines 353-360): "The correlation field (Fig. 13) has two centres of negative correlations – over the North Atlantic and Ural regions with statistically significant correlation coefficients exceeding –0.5. These centres coincide with the locations of the biggest differences in WAF between the ENS+ and ENS– clusters (Fig. 11). The negative correlations indicate that the stronger flux in the regions is associated with weaker stratospheric winds and suggest that errors in the wave activity in the location of the Ural high and Atlantic storm track were crucial for forecasting SSW2018, consistent with the results by Karpechko et al. (2018) and Lee et al. (2019)."

2. Figure 14: For ERA-I, the wave trains associated with MJO phase 6 (contour lines) over the North Pacific and North America, and North Atlantic (Figure AC1) can be seen more clearly in Figure AC1 (including detrending and same period) in comparison to Fig. 14. Furthermore, the anomalies (shaded areas) show a better agreement, especially over North America and the North Atlantic. I suggest to include Figure AC1 instead of Figure 14.

We agree with this suggestion and include Figure AC1 instead of Figure 14 in the revised version of the manuscript and change the time period accordingly in the Data and Methods section.

**Reviewer 2:**

I believe the authors have addressed all of my comments satisfactorily, and if not, I am happy with the response. My one comment is about choosing February 1st as the initialisation date. Currently, I think that the approach in the manuscript is not lucid enough as you cite two papers (Lee et al 2019 and Karpechko et al. 2018 as stating the reasons for choosing this date; lines 99-102). In the response to my query you give a more full answer that would be better placed in the paper. In fact, I think that your figure 1 could be updated to include an extra panel or two (currently it only has one) showing the U at 60N and 10hPa for the 29th January and 5th February reforecast initialisations. The added text to the manuscript would be minimal (~1-2 sentences) but for the reader would be much clearer than simply referring to two previous papers. In all of these three panels for each initialisation date, including the % of the 51 ensemble members that predicted an SSW correctly (or within the +-1 day criterion that they use) would then be useful too.

We thank the reviewer for this suggestion and add two more panels to the Figure 1. The new panels illustrate the prediction skill as the function of lead time and underpin our choice of the ensemble forecast for further analysis. We added the following to the text:

Lines 152-154: In the forecast initialized on 29 January no members showed reversal to easterlies within one day from the observed onset date although four members predicted an SSW to occur in the second half of February (Fig. 1a).

Lines 166-169: With the reduction of the lead time the prediction skill increases rapidly, and all ensemble members of the forecast initialized on 5 February capture the onset of the SSW (Fig. 1c). In this study we focus on the first ensemble forecast predicting the SSW, forecast from 1 February, and contrast behaviour of the members that predicted, and not predicted, the event.

**Reviewer 3:**

The authors have addressed most of my comments. The only minor concern that I still have corresponds to the wave activity flux by Takaya and Nakamura. Following other reviewer's and my suggestion the authors are now showing this flux instead of Plumb flux. However, it would be great if they included the expression as they did when using Plumb flux. It is also important if they could indicate the way the wave-associated fluctuations are computed. For instance, for quasi-stationary they are typically averaged over 5 days.

**We added the formula we used to the Data and Methods section:**

[revised manuscript text omitted]